# Towards Fair Graph Anomaly Detection: Problem, New Datasets, and Evaluation

## Abstract

The Fair Graph Anomaly Detection (`FairGAD`) problem aims to accurately detect anomalous nodes in an input graph while ensuring fairness and avoiding biased predictions against individuals from sensitive subgroups such as gender or political leanings. Fairness in graphs is particularly crucial in anomaly detection areas such as misinformation detection, where decision outcomes can significantly affect individuals. Despite this need, existing works lack realistic datasets that encompass actual graph structures, anomaly labels, and sensitive attributes for research in `FairGAD`. To bridge this gap, we present two novel graph datasets constructed from the globally prominent social media platforms Reddit and Twitter. These datasets comprise 1.2 million and 400,000 edges associated with 9,000 and 47,000 nodes, respectively, and leverage political leanings as sensitive attributes and misinformation spreaders as anomaly labels. We demonstrate that our `FairGAD` datasets significantly differ from the synthetic datasets used currently by the research community. These new datasets offer significant values for `FairGAD` by providing realistic data that captures the intricacies of social networks. Using our datasets, we investigate the performance-fairness trade-off in nine existing GAD and non-graph AD methods on five fairness methods, which sheds light on their effectiveness and limitations in addressing the `FairGAD` problem.

## 1 Introduction

**Background**. Graph Anomaly Detection (GAD) aims to identify anomalous nodes in an input graph whose characteristics are significantly different from those of the rest nodes in the graph (Ding et al., 2019a;b; 2021; Xu et al., 2022). Given that many types of real-world data, including social networks (Kumar et al., 2018; 2019), financial markets (Huang et al., 2022b; Zhang et al., 2023), and cybersecurity (Lakha et al., 2022), can be naturally represented as graphs, there has been an increasing interest in research on developing GAD methods in recent years (Kim et al., 2022; Ma et al., 2021). By detecting anomalies in graphs, we can characterize potential threats and harmful content, enabling early warning, timely intervention, and efficient decision-making. With the advance of Graph Neural Networks (GNNs) (Kipf & Welling, 2017; Liu & Liu, 2021; Liu et al., 2023; Niepert et al., 2016; Zhu et al., 2021), GNN-based GAD methods have increasingly attracted attention in the literature (Kim et al., 2022; Ma et al., 2021).

**Motivation**. Considering fairness in GAD research is essential due to the widespread application of GAD methods in high-stakes domains, such as loan approvals (Huang et al., 2022a) and misinformation (Chen et al., 2022; Wang et al., 2023), where biased and unfair anomaly detection outcomes can have adverse effects on various aspects of our lives (Dong et al., 2021; 2023; Wu et al., 2022). Despite the advances in GAD methods, there has been a notable lack of in-depth investigation into their ability to produce the desired results from a *fairness perspective*. The literature has demonstrated that graph mining algorithms can yield discriminatory results against *sensitive attributes* (*e*.g., gender and political leanings) due to biases introduced/amplified during the mining process (Dong et al., 2023; Kang & Tong, 2021; Wang et al., 2022a). Such observations

Table 1: Statistics of existing relevant datasets and our `FairGAD` datasets. It should be noted that the synthetic graph structure was constructed based on edges formed by structural similarities between nodes (see Agarwal et al. (2021) for the details). That is, our `FairGAD` datasets are new comprehensive benchmark datasets that cover all of graph, anomaly detection, and fairness aspects in the real world.

| Dataset | GAD (Liu et al., 2022b) | | | | | | Fairness (Dai & Wang, 2021; Dong et al., 2022) | | | Fair Non-graph AD (Agarwal et al., 2021) | | | FairGAD | |
|---|---|---|---|---|---|---|---|---|---|---|---|---|---|---|
| | Weibo | Reddit | Disney | Books | Enron | DGraph | Pokec-z | Pokec-n | UCSD34 | German | Credit | Bail | Reddit | Twitter |
| # Nodes | 8,405 | 10,984 | 124 | 1,418 | 13,533 | 3,700,550 | 7,659 | 6,185 | 4,132 | 1,000 | 30,000 | 18,876 | 9,892 | 47,712 |
| # Edges | 407,963 | 168,016 | 335 | 3,695 | 176,987 | 4,300,999 | 29,476 | 21,844 | 108,383 | 24,970 | 2,174,014 | 403,977 | 1,211,748 | 468,697 |
| # Attributes | 400 | 64 | 28 | 21 | 18 | 17 | 59 | 59 | 7 | 27 | 18 | 13 | 385 | 780 |
| Avg. degree | 48.5 | 15.3 | 2.7 | 2.6 | 13.1 | 1.2 | 7.70 | 7.06 | 52.5 | 25.0 | 72.5 | 21.4 | 122.5 | 9.8 |
| Real graph? | ✓ | ✓ | ✓ | ✓ | ✓ | ✓ | ✓ | ✓ | ✓ | ✗ | ✗ | ✗ | ✓ | ✓ |
| Sensitive attributes | - | - | - | - | - | - | Region | Region | Gender | Gender | Age | Race | Political leaning | |
| Attribute bias | - | - | - | - | - | - | 4.3E-4 | 5.4E-4 | 5.3E-4 | 6.33E-3 | 2.46E-3 | 9.5E-4 | 2.22E-3 | 9.14E-4 |
| Structural bias | - | - | - | - | - | - | 8.3E-4 | 1.03E-3 | 6.8E-4 | 1.04E-2 | 4.45E-3 | 1.1E-3 | 4.55E-4 | 6.38E-4 |
| Anomaly labels | Suspicious users | Banned users | Manual label | Tag of amazonfail | Spammer accounts | Overdue accounts | - | - | - | Credit status | Bail decision | Future default | Misinformation spreader | |
| Contamination | 0.103 | 0.033 | 0.048 | 0.020 | 0.004 | 0.004 | - | - | - | 0.300 | 0.221 | 0.376 | 0.137 | 0.067 |
| Correlation | - | - | - | - | - | - | - | - | - | 0.462 | 0.513 | 0.460 | 0.802 | 0.896 |

raise concerns regarding the potential for existing GAD methods to produce *unfair results* in detecting anomalous nodes. However, conducting research on **Fair Graph Anomaly Detection (`FairGAD`)** is quite challenging, primarily due to the *absence of comprehensive benchmark datasets* that encompass all of the graph, anomaly detection, and fairness aspects. As depicted in Table 1, the existing datasets for fairness or anomaly detection research have synthetic graph structures, or lack anomaly labels or sensitive attributes. The use of such synthetic data fails to reflect real-world properties, while the lack of anomaly labels or sensitive attributes prevents the reasonable evaluation of existing GAD methods from a fairness perspective.

**Our Work**. We create two datasets that have real graph structure, real anomaly labels, and real sensitive attributes, and evaluate existing GAD methods in terms of both performance and fairness by using the newly created datasets. The contributions of our work are: (1) **Problem Formulation:** We define the `FairGAD` problem, which serves as the foundation of our investigation regarding fairness in GAD research; (2) **Novel Datasets:** We create two datasets from two major social media platforms, *i.e.*, Twitter and Reddit, and analyze their crucial properties such as contamination and attribute/structural biases; and (3) **Experimental Evaluation:** Under the `FairGAD` problem, using our datasets, we examine the effectiveness of four state-of-the-art GAD methods (Ding et al., 2019a; Xu et al., 2022; Liu et al., 2021; Huang et al., 2023) and seven non-graph AD methods (Bandyopadhyay et al., 2020; Li et al., 2022; Kingma & Welling, 2014; Breunig et al., 2000; Liu et al., 2008; Bandyopadhyay et al., 2019) in terms of accuracy and fairness as a solution to the `FairGAD` problem. Additionally, we explore the impact of incorporating five fairness methods (Shekhar et al., 2021; Zeng et al., 2021; Dong et al., 2022; Rahman et al., 2019) into the GAD methods. Our findings suggest that they fail to produce the desired outcomes, highlighting the need for further investigation and follow-up studies on `FairGAD`. For **Reproducibility**, our code and datasets are available at Anonymous GitHub.

## 2 THE PROPOSED PROBLEM: `FAIRGAD`

**Problem Definition.** The GAD problem is commonly approached as an unsupervised node classification task on a graph (*a.k.a.*, network), aiming to determine whether nodes in the graph are anomalies (*a.k.a.*, outliers) or not (Ding et al., 2019a; Liu et al., 2021; Xu et al., 2022). Anomalies typically consist of a minority of the nodes in the graph. Let $\mathcal{G} = (\mathcal{V}, \mathcal{E}, \mathbf{X})$ represent an attributed graph, where $\mathcal{V}$ and $\mathcal{E}$ denote the sets of nodes and edges, respectively, and $\mathbf{X} \in \mathbb{R}^{n \times d}$ represents the node feature matrix, where $n$ indicates the number of nodes in the graph and $d$ indicates the number of attributes for each node. The adjacency matrix is denoted by

$\mathbf{A} \in \{0, 1\}^{n \times n}$. The anomaly labels are represented as $\mathbf{Y} \in \{0, 1\}^n$, where a value of 1 indicates that the node is an anomaly, and the predictions of the model are denoted as $\hat{\mathbf{Y}}$. GAD methods aim to identify the nodes whose patterns differ significantly from the majority in terms of both attributes and a structure. It is worth noting that since GAD is regarded as an unsupervised problem in most literature (Kim et al., 2022; Ma et al., 2021), the labels should only be used in the test step, not in the training step.

The `FairGAD` problem extends beyond GAD by incorporating *sensitive attributes* for nodes. In a social network, features such as age and gender, which users are usually reluctant to share, are considered sensitive attributes. Thus, one of the features for each node should include a sensitive attribute, which can be represented as $\mathbf{S} \in \{0, 1\}^n$ if the attribute is binary – one having a sensitive attribute of 0 (*e*.g., male) and the other having a sensitive attribute of 1 (*e*.g., female). `FairGAD` methods aim to accurately detect anomalous nodes while avoiding discriminatory predictions against individuals from any specific sensitive group.

**Metrics.** We employ two types of metrics. *Performance metrics* are used to evaluate the accuracy of the GAD method while considering the imbalanced ratio between anomaly and normal nodes. For this purpose, the Area Under the ROC Curve (AUCROC) is widely utilized in the literature (Dong et al., 2022; Liu et al., 2022b; 2021). Additionally, we employ the Area Under the Precision-Recall Curve (AUPRC), which is more sensitive to minority labels and thus suitable for GAD. Higher values of these metrics indicate better model performance. *Unfairness metrics* are used to evaluate the fairness of the GAD methods when predicting anomalies with respect to the node's sensitive attribute. Statistical Parity (SP) (Agarwal et al., 2021; Beutel et al., 2017; Louizos et al., 2016) measures the difference in prediction rates for anomalies across the two node groups with different sensitive attributes, *i*.e., $SP = |P(\hat{\mathbf{Y}} = 1|\mathbf{S} = 0) - P(\hat{\mathbf{Y}} = 1|\mathbf{S} = 1)|$. Another fairness measure is the Equality of Odds (EOO) (Agarwal et al., 2021; Beutel et al., 2017; Louizos et al., 2016), which quantifies the difference in true positive rates of the method when detecting anomalies across different sensitive attributes, *i*.e., $EOO = |P(\hat{\mathbf{Y}} = 1|\mathbf{S} = 0, \mathbf{Y} = 1) - P(\hat{\mathbf{Y}} = 1|\mathbf{S} = 1, \mathbf{Y} = 1)|$. Lower values of fairness metrics indicate better fairness achieved by the method.

## 3 DATA DESCRIPTION

**Collection Procedure.** In this study, we focus our analysis on two globally-prominent social media platforms, namely Twitter and Reddit. Both Twitter and Reddit exemplify large-scale, mainstream social media platforms, which boast substantial user engagement levels and global penetration, and are among the top 10 most visited websites in the world (Semrush, 2023; Wikipedia, 2023). Another essential reason for selecting these platforms lies in their broad utilization within previous studies (Jin et al., 2022; Kang et al., 2022; Kumar et al., 2019; Ma et al., 2023; Shu et al., 2019; Verma et al., 2022; Yang et al., 2022). Given the significant user base and the wide array of communication exchanges that occur on these platforms, they have been consistently employed as fertile grounds for research across various domains, including political discourse (Valiavska & Smith-Frigerio, 2022) and information propagation (Gomez Rodriguez et al., 2013; Keegan, 2019; Ng et al., 2022), among others. However, as highlighted in Section 1, there have been no studies that investigate the interplay between graph structure, anomaly detection, and fairness due to the scarcity of realistic datasets, including on the Reddit and Twitter platforms. Recognizing the significance of the `FairGAD` problem as well as the widespread use of these two social platforms, we create two representative datasets derived from Twitter and Reddit to address this gap in research.

**Dataset Curation.** For the **Twitter** dataset, we used all historical posts, user profiles, and the follower relationships of 47,712 Twitter users using the Twitter API. The list of Twitter users we collected were derived from Verma et al. (2022), who posted COVID-19 related tweets that contain misinformation. For the **Reddit** dataset, we first identified a list of 110 politics-related subreddits (shown in Appendix B). Then, we used the Pushshift API to collect all historical posts of these subreddits and identified those users who had participated in the discussions of these subreddits. Finally, we randomly sampled users from the participants of these

subreddits and collected all of their historical posts since their account creation. The collection of publicly available datasets was determined to be review exempt by the Institutional Review Board (IRB).

In both datasets, the **political leaning of users** is defined as the **sensitive attribute**. The **anomaly label** represents whether a user is a **real-news or misinformation spreader**[1]. We note that the correlation between political leanings and the spread of misinformation is a well-known social phenomenon that has been firmly established in several prior studies (Grinberg et al., 2019; Cohen et al., 2020; Lawson & Kakkar, 2022; Gupta et al., 2023). To classify these labels, we leverage the FACTOID dataset (Sakketou et al., 2022), which provides the two lists of online news outlet domains corresponding to 1,577 misinformation and 571 real news sources, and 142 left-leaning and 777 right-leaning domains. We followed the same strategy to categorize hyperlinks based on their domains, classifying them as left/right leaning as well as real news/misinformation. Consequently, users are assigned a sensitive attribute of 1 if they post a higher number of links from right-leaning sites than left-leaning sites, and 0 in the opposite case. Similarly, users are assigned an anomaly label value of 1 if they post a greater number of misinformation links than real news links.

Furthermore, we created the **graph structure** in both datasets. For Reddit, the graph was constructed by linking two users who posted to the same subreddit within a 24-hour window. This creates an undirected edge between the users, reflecting the non-hierarchical nature of interactions within the subreddit. This design decision was inspired by prior research indicating that users who interact within the same online community in close temporal proximity are likely to be aware of each other's posts or share similar topical interests (Krohn & Weninger, 2022; Waller & Anderson, 2019; Weerasinghe et al., 2022). We employed Sentence Transformers (Reimers & Gurevych, 2019) to generate embeddings from the users' post histories. We then took the average of a user's post embeddings and aggregate it with their sensitive attribute to derive the node feature in our graph. In the case of Twitter, our approach was slightly different due to the platform's distinct user interaction mechanics. A directed edge is created from user $A$ to user $B$ if user $A$ follows user $B$. To incorporate node features, we inferred user demographic information by using the M3 System (Wang et al., 2019b), a multimodal, multilingual, and multi-attribute demographic inference framework trained on massive Twitter data. By doing so, we obtained the age group ($\leq 18$, 19-29, 30-39, $\geq 40$), gender, and whether the Twitter account is an organization account based on the user profile and historical tweets. We also obtained the number of favorites, and the status of each account if it was verified. Users' post histories were also retrieved and embedded using a multilingual model (Reimers & Gurevych, 2020), and the average of a user's post embeddings was concatenated with the above user information to form the node features. For both datasets, we took the largest connected component of nodes as final graph structures.

Addressing concerns about fairness in politically biased misinformation detection has significant practical implications. Political bias can result in problems such as the reinforcement of confirmation biases, unequal treatment of news sources, and hindrances in achieving just and precise categorization. In this case, misclassifying minority groups as misinformation spreaders can amplify biases and stereotypes and potentially reinforce existing divisions through algorithmic misuse. Prioritizing fairness builds trust in the detection process and fosters a more equitable information environment. This strategy not only alleviates unintentional consequences, but also facilitates principled and ethical application of misinformation detection systems.

**Dataset Statistics.** Table 1 provides an overview of the basic statistics and key properties such as correlation, attribute/structural bias, and contamination: (1) **correlation** indicates the correlation coefficient between sensitive attributes and anomaly labels; (2) **attribute bias** (Dong et al., 2022) employs the Wasserstein-1 distance (Villani, 2021) to compare the distribution of node attributes between anomalies and non-anomalies; (3) **structural bias** (Dong et al., 2022) uses the Wasserstein-1 distance (Villani, 2021) while comparing adjacency matrices based on a two-hop neighborhood between them; and (4) **contamination** represents the proportion of anomaly nodes in the dataset. Appendix C provides the equations to quantify these properties.

---

[1]The term "misinformation" is used in a political context and serves as an overarching categorization that encompasses several dimensions (Sakketou et al., 2022).

**Discussions.** We summarize the key differences between `FairGAD` and the synthetic datasets, *i.*e., German, Credit, and Bail. First, our datasets show a strong link between sensitive attributes and anomalies. This supports previous studies (Grinberg et al., 2019; Gupta et al., 2023; Lawson & Kakkar, 2022) on the correlation between political leanings and the spread of misinformation. As a result, a naive approach to infer anomalies based on the sensitive attributes of nodes could result in high accuracy in our datasets. However, this implies that the approach harms fairness by preserving the inherent correlations. Furthermore, since such correlations in a dataset can leak into the graph structure and non-sensitive attributes (Dong et al., 2022; Wang et al., 2022b), GAD methods have the potential to amplify the aforementioned biases.

In addition, our datasets present different graph structures that are shaped by the features of social media platforms. On Reddit, users participate in many different subreddits, resulting in a denser graph compared to the synthetic ones. In contrast, Twitter has a sparser graph than the synthetic ones due to its directed edges that represent user-following relationships, leading to a lower average degree.

The synthetic datasets were first created for non-graph AD, where synthetic edges were formed by linking nodes using the Minkowski distance, without any consideration of actual user behavior. It implies that the inductive biases of GAD methods may not be as applicable since they often rely on assumptions that anomalies differ from their neighboring nodes (Liu et al., 2021; Xu et al., 2022). In contrast, our datasets exhibit a lower degree of structural bias compared to the synthetic ones due to its origins in actual user behavior. This difference is because users with distinct properties may still be connected in social networks. This is supported by the average similarity between users connected by edges of 43% and 44% for our Reddit and Twitter, respectively, which contrasts with the thresholds used to create the synthetic edges for German, Credit, and Bail at 80%, 70%, and 60%, respectively (Dong et al., 2022).

Lastly, our Twitter dataset exhibits the lowest attribute bias out of our `FairGAD` and the synthetic datasets. Additionally, it includes a larger number of attributes than other datasets. According to Zimek et al. (2012), such properties (*i.*e., low attribute bias and high dimensionality of attributes) are known to make anomaly detection more challenging, which will be demonstrated in Section 4.2.

# 4 EVALUATION

## 4.1 EXPERIMENTAL SETTINGS

**GAD Methods.** We employ *three GAD methods*, *i.*e., DOMINANT (Ding et al., 2019a), CONAD (Xu et al., 2022), and CoLA (Liu et al., 2021). Our goal is to present new datasets and to investigate their properties and applicability in terms of graphs, fairness, and anomaly detection aspects. Therefore, we have chosen representative or state-of-the-art GAD methods rather than using all GAD methods. In addition, we will examine *a recent GAD method* (*i.*e., VGOD (Huang et al., 2023)), *five non-graph AD methods* (*i.*e., DONE (Bandyopadhyay et al., 2020), AdONE (Bandyopadhyay et al., 2020), ECOD (Li et al., 2022), VAE (Kingma & Welling, 2014), and ONE (Bandyopadhyay et al., 2019)), and *two heuristic methods* (*i.*e., LOF (Breunig et al., 2000) and IF (Liu et al., 2008)) in Appendix I.

DOMINANT (Ding et al., 2019a) uses GCNs to obtain node embeddings, which are then used in other GCNs to reconstruct the attribute and the adjacency matrices. By measuring the errors between the original and decoded matrices, anomalies are detected. Under the premise that anomalous nodes are more difficult to encode than normal nodes, it ranks nodes based on their reconstruction errors. The top nodes with high reconstruction errors are identified as anomalies. CONAD (Xu et al., 2022) incorporates human knowledge about different anomaly types into detecting anomalies through knowledge modeling. Synthetic anomalies are introduced into the graph for self-supervised learning via a contrastive learning loss. The reconstruction error is then used to label nodes as anomalies. CoLA (Liu et al., 2021) employs self-supervised learning with pairs of a contrastive node and local neighborhood obtained by random walks. This subsampling strategy assumes that anomalies and their neighborhoods differ from normal nodes and their neighborhoods. The

learned model compares all nodes in the graph with their neighborhoods via positive and negative pairs to identify nodes, which are then predicted as anomalies.

**Fairness Methods.** We employ *five fairness methods* that are applicable to the GAD problem: (1) fairness regularizers: FAIROD (Shekhar et al., 2021), CORRELATION (Shekhar et al., 2021), and HIN (Zeng et al., 2021); (2) graph debiasers: EDITS (Dong et al., 2022) and FAIRWALK (Rahman et al., 2019). Detailed equations for these methods can be found in Appendix D.

*Fairness Regularizers.* FAIROD (Shekhar et al., 2021) introduces two losses $\mathcal{L}_{FairOD}$ for SP, which reduces the sum of reconstruction errors, and $\mathcal{L}_{ADCG}$ for EOO, which penalizes the fair model for ranking nodes differently from the original model. The CORRELATION regularizer $\mathcal{L}_{Corr}$, derived from the FAIROD implementation, measures the correlation between sensitive attributes and node representation errors by using the cosine rule. The HIN (Zeng et al., 2021) regularizer $\mathcal{L}_{HIN}$ penalizes the difference in prediction rates between sensitive attribute groups for both anomalies and non-anomalies. Zeng et al. (2021) introduces another function that reduces EOO, but requires labels. Thus, we use $\mathcal{L}_{ADCG}$ from the FAIROD regularizer as a replacement. To incorporate the fairness regularizer methods (*i.e.*, FAIROD, CORRELATION, and HIN) into the GAD methods, we made the following modifications to the GAD methods: (1) $\mathcal{L} = \mathcal{L}_o + \lambda\mathcal{L}_{FairOD} + \gamma\mathcal{L}_{ADCG}$ for FAIROD; (2) $\mathcal{L} = \mathcal{L}_o + \lambda\mathcal{L}_{HIN} + \gamma\mathcal{L}_{ADCG}$ for HIN; and (3) $\mathcal{L} = \mathcal{L}_o + \lambda\mathcal{L}_{corr}$ for CORRELATION, where $\mathcal{L}_o$ denotes the original loss of the GAD method, and $\lambda$ and $\gamma$ are hyperparameters.

*Graph Debiasers.* EDITS (Dong et al., 2022) takes the graph and node features as input and employs gradient descent to learn a function that debiases them by reducing the estimated Wasserstein distance between attribute dimensions and the node label. This results in modifications to the adjacency matrix (by removing or adding edges) as well as the node feature matrix, while keeping the node labels unchanged. FAIRWALK (Rahman et al., 2019) aims to generate fairer node embeddings of a graph without relying on node features, only using sensitive attributes. It modifies random walks in the graph by considering the sensitive attribute of the nodes at each step of the random walk. This ensures that the nodes with a minority sensitive attribute are explored more, leading to fairer representations. We use these embeddings as node features for GAD methods.

## 4.2 RESULTS AND ANALYSIS

Due to space limitations, we provide additional interesting results on alternative sensitive attributes such as age and gender, and human verification for the classification of labels in Appendix G and H, respectively.

**Using GAD Methods (without Fairness Methods).** We evaluate the performance and fairness of GAD methods without incorporating any fairness methods, as shown in the 'without fairness methods' columns (*i.e.*, ×) in Table 2. In general, the accuracy (*i.e.*, AUCROC and AUPRC) of GAD methods on Reddit tends to be higher than their accuracy on Twitter. However, we found the suboptimal performance of existing GAD methods in terms of accuracy, which may be influenced by several factors. One possible reason is that our datasets manifest less structural bias than existing synthetic datasets, which may result in the limited performance of GAD methods due to their prevalent reliance on graph homophily. Furthermore, we observe that striving for higher accuracy via existing GAD methods adversely affects their fairness, which leads to higher SP and EOO. That is, they show worse SP and EOO on Reddit than on Twitter. Considering that the attribute bias of Reddit is significantly larger than that of Twitter while their structural biases are similar (see Table 1), we attribute the results of high SP and EOO on Reddit to its substantial attribute bias.

**Impact of Graph Debiasers (FAIRWALK and EDITS).** We investigate the impact of debiased graphs and node embeddings obtained through FAIRWALK and EDITS, respectively. Interestingly, we observe that the debiased graph from **EDITS** leads to a noticeable improvement in the accuracy of the GAD methods, while their unfairness escalates, as indicated by larger values for SP (except for COLA) and EOO. This observation contradicts the claim made in Dong et al. (2022) that EDITS can reduce unfairness while maintaining accuracy. We speculate that this discrepancy arises from the attribute debiaser used in EDITS, which

Table 2: Performance and fairness results of GAD methods on our original and debiased datasets. We encountered out-of-memory (*i.e.*, 'o.o.m') when attempting to obtain the debiased Reddit graph from EDITS.

**(a) Twitter Dataset**

| Methods | CoLA | | | CONAD | | | DOMINANT | | |
|---|---|---|---|---|---|---|---|---|---|
| Debiasers | × | EDITS | FairWalk | × | EDITS | FairWalk | × | EDITS | FairWalk |
| AUCROC (↑) | 0.443±0.006 | 0.452±0.013 | **0.488±0.006** | 0.558±0.007 | **0.704±0.001** | 0.536±0.009 | 0.560±0.007 | **0.704±0.001** | 0.535±0.009 |
| AUPRC (↑) | 0.052±0.001 | 0.053±0.002 | **0.062±0.002** | 0.087±0.001 | **0.173±0.001** | 0.085±0.005 | 0.088±0.001 | **0.173±0.001** | 0.085±0.005 |
| SP (↓) | 0.028±0.003 | **0.007±0.006** | 0.008±0.006 | 0.038±0.006 | 0.289±0.004 | **0.011±0.004** | 0.040±0.006 | 0.289±0.003 | **0.012±0.004** |
| EOO (↓) | 0.023±0.012 | 0.009±0.005 | **0.001±0.001** | 0.044±0.003 | 0.278±0.004 | **0.013±0.002** | 0.044±0.003 | 0.278±0.003 | **0.013±0.002** |

**(b) Reddit Dataset**

| Methods | CoLA | | | CONAD | | | DOMINANT | | |
|---|---|---|---|---|---|---|---|---|---|
| Debiasers | × | EDITS | FairWalk | × | EDITS | FairWalk | × | EDITS | FairWalk |
| AUCROC (↑) | 0.453±0.014 | o.o.m | **0.502±0.004** | 0.608±0.001 | o.o.m | 0.517±0.024 | **0.608±0.001** | o.o.m | 0.518±0.023 |
| AUPRC (↑) | 0.032±0.018 | o.o.m | **0.140±0.005** | 0.200±0.001 | o.o.m | 0.149±0.015 | **0.200±0.001** | o.o.m | 0.150±0.016 |
| SP (↓) | 0.035±0.027 | o.o.m | **0.006±0.004** | 0.132±0.001 | o.o.m | **0.025±0.017** | 0.133±0.002 | o.o.m | **0.021±0.015** |
| EOO (↓) | 0.177±0.014 | o.o.m | **0.003±0.003** | 0.055±0.002 | o.o.m | **0.028±0.018** | 0.057±0.003 | o.o.m | **0.025±0.017** |

focuses on minimizing the difference in node attribute distributions as a whole, not just the node distributions with respect to the sensitive attribute. Furthermore, we observed that EDITS significantly enhance the accuracy of CONAD and DOMINANT, which fully exploits the augmented graph structure based on the reconstruction error. However, the accuracy of CoLA achieved only a minor improvement since it partially exploits the augmented graph structure by sampling node pairs through random walks. On the other hand, the modifications made by **FAIRWALK** consistently and remarkably improve fairness, demonstrated by the decrease in both SP and EOO. In terms of accuracy, the GAD methods show different trends whether they use the reconstruction error in the attribute matrix. Specifically, the accuracy of DOMINANT and CONAD, which jointly learn the reconstruction errors in the adjacency and attribute matrices, decreases, while the accuracy of CoLA, which solely relies on the graph structure, increases. As mentioned in Section 4.1, we use the FAIRWALK embeddings instead of node attributes as node features to reduce the attribute bias. Thus, the optimization of DOMINANT and CONAD becomes more challenging without the use of node attributes.

**Impact of Fairness Regularizers (FAIROD, HIN, and CORRELATION).** As mentioned in Section 4.1, we add the regularizers to the original loss function of each GAD method. Since different regularizers require different weight scales (*i.e.*, $\lambda$ and $\gamma$), we perform separate hyperparameter grid searches for each regularizer.

*FairOD and HIN*. We investigate how AUCROC (*i.e.*, performance) and EOO (*i.e.*, fairness) vary with changes in the weight of $\lambda$ for $\mathcal{L}_{FairOD}$, $\mathcal{L}_{HIN}$ and $\gamma$ for $\mathcal{L}_{ADCG}$. Due to space limitations, we here report the results of CONAD since we confirmed that the results of other methods are consistent with those of CONAD. Instead, we provide the omitted results in Appendix E. Figure 1 illustrates the results of CONAD using HIN and FAIROD with varying $\lambda$ and $\gamma$ values on Reddit. Regarding the **performance** metric, where a higher value is better, we observe that increasing $\lambda$ continuously leads to a decrease in AUCROC, and the magnitude of the AUCROC decrease increases as $\lambda$ increases. On the other hand, smaller $\gamma$ values result in improved performance, while larger $\gamma$ values lead to decreased performance as CONAD fails to minimize its original representation loss. Regarding the **fairness** metric, on Reddit, there is a general trend that increasing $\gamma$ tends to result in a decrease in EOO, except for the cases of CONAD with the FAIROD and HIN regularizers. Conversely, the impact of increasing $\lambda$ varies depending on the specific combinations of GAD methods and regularizers. For instance, in the combinations of DOMINANT and FAIROD, as well as CONAD and FAIROD, increasing $\lambda$ leads to a decrease in EOO. However, in other combinations, there appears to be no significant correlation between $\lambda$ and EOO. This inconsistency in results becomes more pronounced when examining Twitter. We conjecture that these differences are due to the inherent complexity of addressing fairness considerations, particularly given the relatively large attribute bias inherent on Twitter. Therefore, the results indicate that we can achieve improvements in both performance and fairness by appropriately setting the values of $\gamma$. However, we believe that the gain of improvement is not substantial in either metric.

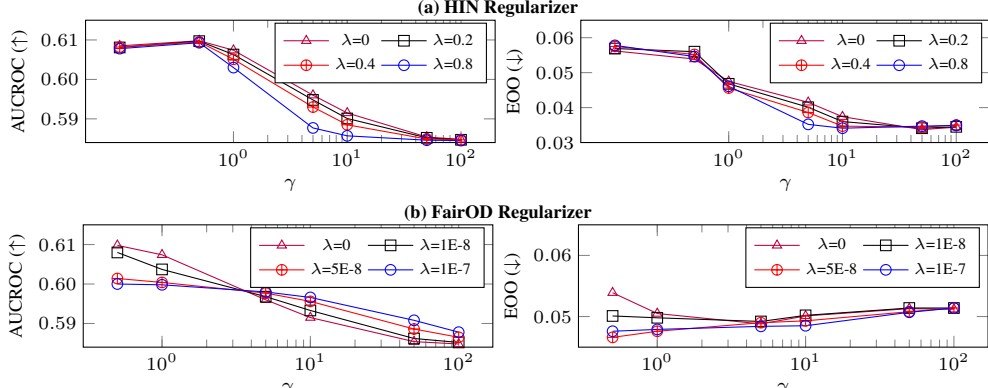

Figure 1: Changes in AUCROC (left) and EOO (right) for different values of $\lambda$ (HIN or FairOD factor) and $\gamma$ (ADCG factor) for CONAD method with HIN and FAIROD regularizers on Reddit.

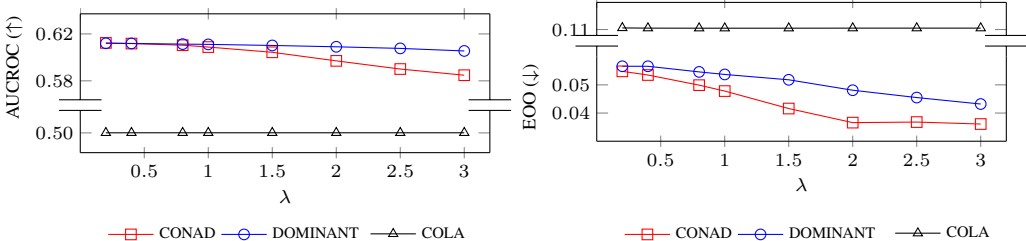

Figure 2: Changes in AUCROC (left) and EOO (right) for different values of $\lambda$ (Correlation factor) for CONAD, DOMINANT, and CoLA methods with CORRELATION regularizer on Reddit.

*Correlation.* Since CORRELATION only requires a single weight parameter $\lambda$, we present the results of all GAD methods using CORRELATION in Figure 2. Except for CoLA, the results show that increasing $\lambda$ consistently leads to lower EOO. However, the magnitude of the performance drop by increasing $\lambda$ varies across methods. We note the differences between the original losses of CONAD and DOMINANT; DOMINANT simply uses the node reconstruction error to rank anomalies, while CONAD is trained on augmented graphs that encode known anomaly types in addition to the node reconstruction error. As such, modifying the joint loss of CONAD would have a more significant impact on its learning.

*CoLA with Fairness Regularizers.* While the fairness regularizers consider reconstruction errors in their formulations, CoLA relies on the differences between positive and negative neighbor pairs. For this reason, the fairness regularizers do not directly contribute to the learning mechanism in CoLA. When fairness regularizers are introduced, the results of CoLA exhibit a significant standard deviation, since only emphasizing the losses of certain nodes may not always result in improved performance or fairness. This observation highlights the need to develop alternative fairness regularizers that can be effectively incorporated into GAD models with different mechanisms other than reconstruction error.

*Accuracy-Fairness Trade-off Space.* We present the trade-off space between accuracy and fairness across CONAD and DOMINANT with the fairness regularizers in Figure 3; the results for CoLA can be found in Appendix E. Note that an ideal `FairGAD` method should achieve high AUCROC and low EOO, which would position it in the bottom right corner in Figure 3. However, most GAD methods, even after applying the fairness methods, lie along a straight line, indicating a linear trade-off between performance and fairness. The trade-off space under the FAIROD regularizer appears to be worse than that under the HIN and CORRELATION regularizers. For FAIROD, the space exhibits a tendency to deviate considerably from the optimal placement

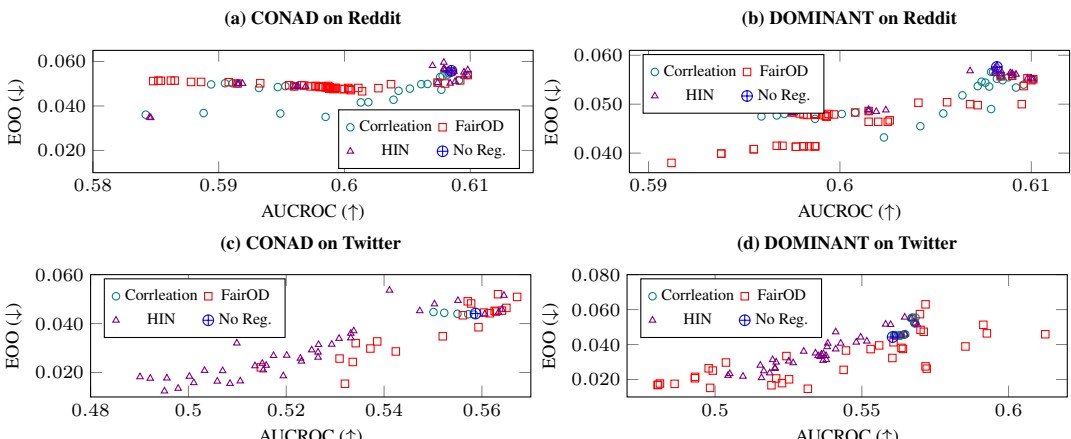

Figure 3: Trade-off spaces for CONAD (left) and DOMINANT (right) with fairness regularizers. The ideal FairGAD method should have low EOO and high AUCROC (*i.e.*, the bottom right corner).

in the bottom right corner and displays a significant distance between instances. This could be attributed to its formulation, which includes the sum and standard deviation of reconstruction errors without a direct link to the sensitive attribute. On the other hand, HIN and CORRELATION penalize the method for having a large difference in errors between sensitive attribute groups. Considering that most GAD methods rely on reconstruction errors to detect anomalies, the formulation of HIN and CORRELATION helps to improve the trade-off space to some extent. Nevertheless, none of the existing GAD methods achieve the desired outcomes (*i.e.*, bottom right corner). This means that it is currently difficult to detect misinformation among right-leaning users. As a result, political bias in FairGAD can lead to problems such as reinforcement of confirmation bias, unequal treatment of news sources, and difficulties in achieving fair and accurate categorization.

## 5 CONCLUSION

In this work, we defined an important yet under-explored problem, namely `FairGAD`, and presented two novel and realistic `FairGAD` datasets that cover aspects of the graph, anomaly labels, and sensitive attributes. Through extensive experiments, we demonstrate that incorporating existing fairness methods into existing GAD methods does not yield the desired outcomes, indicating a linear trade-off between performance and fairness. This finding emphasizes the need for further investigations and follow-up studies on `FairGAD`.

**Limitations and Future Work.** We defined the `FairGAD` problem as an unsupervised node classification task. For a fair comparison, we employed DOMINANT (Ding et al., 2019a), CONAD (Xu et al., 2022), and CoLA (Liu et al., 2021) based on unsupervised learning. However, a few GAD methods (Liu et al., 2022c; Wang et al., 2019a) devised semi-supervised learning that leverages a limited subset of labels. Thus, further studies can explore the impact of semi-supervised learning on the accuracy and fairness of GAD methods. In such cases, the only extra tasks are to analyze the parameters for the labels given to the model as well as the ratio of labels pertaining to sensitive attributes. Another potential approach to improve the trade-off between accuracy and fairness is to combine both graph debiasers and fairness regularizers. In Appendix F, we present initial findings demonstrating how this integration investigates a unique portion of the trade-off space compared to using regularizers alone. However, a comprehensive analysis can investigate the impact that such combinations have and their suitability for various GAD methods. Lastly, by infusing this temporal aspect, our datasets will be better able to accommodate the evolution of our problem beyond the confines of a static framework. For instance, we can shed light on critical junctures where fairness considerations may become more pronounced in the context of evolving graph structures.

## 6 ETHICS STATEMENT

**User Privacy.** We clarify that our datasets for release are created only from public data available from the Pushshift dataset (refer to `https://github.com/pushshift/api`) for Reddit data and Verma et al. (2022) for Twitter data. To protect user privacy, our datasets do not include any private information about users, such as the actual user names or IDs. In particular, each user's postings were encoded into a low-dimensional embedding vector, thus they do not contain the raw text of users' posts. Therefore, we cannot identify specific users from our datasets nor can we infer an actual user's political leanings.

We will follow the procedures documented in previous studies in data mining and social computing (Ao et al., 2021; Beel et al., 2022; Jin et al., 2023; Kumar et al., 2018; Saveski et al., 2022; Wu et al., 2020) to further protect user privacy when releasing the data. We will also prepare a Data Use Agreement for other researchers to sign in order to gain access to the datasets, and a point of contact for users to inquire if they are part of the dataset and to remove their information upon request.

**Risk of Perpetuating Biases.** The correlation between political leanings and the spread of misinformation is a well-known social phenomenon that has been firmly established in several prior studies (Grinberg et al., 2019; Gupta et al., 2023; Lawson & Kakkar, 2022). However, the intention behind this study is NOT to suggest a direct link between political leanings and misinformation propagation, NOR to perpetuate any stereotypes or biases that might result from such a link. Instead, our study has two fundamental objectives. One is to examine whether this correlation actually exists in our datasets, which include real-world user behaviors collected from globally-prominent social media platforms. Another is to investigate whether existing GAD methods yield biased outcomes on our datasets due to these inherent biases, and whether existing fairness methods can be effectively incorporated into GAD methods to produce fairer results.

**Data Reconstruction.** Reconstructing the original data by using users' low-dimensional embeddings is infeasible in practice due to the requirement for uniform settings across input formats (*e*.g., the length of the raw text and profile images), model parameters (*e*.g., Sentence Transformers and M3 system), and more. Notably, user demographic information is inferred via the M3 system, not directly provided by users. Thus, reconstructing the demographic information of a user may not necessarily correspond to the actual identification of the associated user.

**Bot Accounts.** For the Twitter dataset, we have used the Botometer[2] to detect bot users with a threshold of 0.9, resulting in about 1,164 users (*i*.e., 3.8% of all users) being labeled as bot accounts. We will make the lists of bot accounts readily detected available to researchers in the form of node labels, as we do not store usernames in our dataset. Meanwhile, we found that only 35 bot accounts were classified as misinformation spreaders out of the total number of detected bot accounts. The results demonstrate that misinformation spreaders and bot accounts are a distinct categorization. Unfortunately, for the Reddit dataset, there is a lack of well-established methods for detecting bot accounts. Therefore, we acknowledge the potential inclusion of bot accounts in the Reddit dataset.

**Representativeness of Our Datasets.** We would like to clearly address the scope of our datasets, specifically with respect to users engaging in discussions about COVID-19 (for the Twitter dataset) and politically-related misinformation (for the Reddit dataset). As such, it is important to recognize that our datasets do not fully represent the broader populations on Twitter and Reddit. The rationale for selecting these particular topics is based on the extensive research exploring misinformation propagation in the context of COVID-19 and politics (Bin Naeem & Kamel Boulos, 2021; He et al., 2021; Lu et al., 2023; Micallef et al., 2020): (Politics) this is particularly significant given the potential polarization and ideological divisions that may arise from the spread of such misinformation, which can affect public discourse and decision-making; (COVID-19) unverified claims or inaccurate information about the virus, prevention methods, and treatments can lead to misguided actions that exacerbate the impact of the pandemic and hinder effective response efforts.

---

[2]`https://botometer.osome.iu.edu/`

## 7 REPRODUCIBILITY STATEMENT

All the experiments were conducted with the NVIDIA DGX-1 system with 8 NVIDIA TESLA V100 GPUs. Each experiment was repeated twenty times to ensure the robustness and reliability of the results. We used the PyGOD[3] (Liu et al., 2022a) implementation of DOMINANT, CONAD, and COLA methods. Our code and datasets are available at Anonymous GitHub. We plan to release all the code and datasets upon the acceptance of this paper. For the full reproducibility, we provide complete implementation details in Appendix K.

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

# Towards Fair Graph Anomaly Detection:
# Problem, New Datasets, and Evaluation (Appendix)

## A DATASET DOCUMENTATION

We provide the dataset documentation by following the guidelines of the Datasheets for Datasets Gebru et al. (2021). Our code and datasets are available at Anonymous GitHub. We plan to release all the code and datasets upon the acceptance of this paper.

### A.1 MOTIVATION

**For what purpose was the dataset created? Was there a specific task in mind? Was there a specific gap that needed to be filled?**
The datasets were created to study the `FairGAD` problem, which aims to accurately detect anomalous nodes in an input graph while avoiding biased predictions against individuals from sensitive subgroups. To the best of our knowledge, we are the first to present comprehensive and real-world ("organic") benchmark datasets that cover all of the graph, anomaly detection, and fairness aspects.

### A.2 COMPOSITION

**What do the instances that comprise the dataset represent (*e.g.*, documents, photos, people, countries)? Are there multiple types of instances (*e.g.*, movies, users, and ratings; people and interactions between them; nodes and edges)?**
Each node in our `FairGAD` datasets represents a user who had posts on the Reddit or Twitter platforms. Each undirected edge in the Reddit dataset is created between two users if they post to the same subreddit within a 24-hour window, while each directed edge from user $A$ to user $B$ in the Twitter dataset represents user $A$ following user $B$.

**How many instances are there in total (of each type, if appropriate)?**
As shown in Table 1, there are 9,892 and 47,712 nodes, and 1,211,748 and 468,697 edges in the Reddit and Twitter datasets, respectively.

**Does the dataset contain all possible instances or is it a sample (not necessarily random) of instances from a larger set? If the dataset is a sample, then what is the larger set? Is the sample representative of the larger set (*e.g.*, geographic coverage)? If so, please describe how this representativeness was validated/verified. If it is not representative of the larger set, please describe why not (*e.g.*, to cover a more diverse range of instances, because instances were withheld or unavailable).**
The datasets used in our study comprise the largest connected component of each graph, consisting of all crawled users. For the Reddit dataset, we collect a list of active users in political subreddits. Here, we use a crowd-sourced collection of political subreddits[4] (see Appendix B) following previous works (Nithyanand et al., 2017). These users were collected from the period of December 10th, 2022 until March 30th, 2023. It is important to note that these users may not reflect the overall Reddit user population.

Regarding the Twitter dataset, the users were obtained from a study conducted by Verma et al. (2022), which focused on analyzing COVID-19 misinformation shared among Twitter users. Consequently, the dataset is representative of users discussing misinformation related to COVID-19 on Twitter and thus may not be representative of the entire Twitter population.

---

[4]https://www.reddit.com/r/redditlists/comments/josdr/list_of_political_subreddits/

By utilizing these datasets, we aim to gain insights into the `FairGAD` problem within the specific contexts of political leanings and misinformation spreaders on Reddit and Twitter, shedding light on the interplay between graph structures, anomaly detection, and fairness within these platforms.

**What data does each instance consist of? "Raw" data (*e.g.*, unprocessed text or images) or features? In either case, please provide a description.**
Each user in the Reddit dataset is associated with data reflecting their political leanings, which was determined using the methodology outlined in Sakketou et al. (2022). Additionally, we used an average embedding of each user's all posts made in the corresponding political subreddits using the `all-MiniLM-L6-v2` model provided by Sentence-Transformers[5].

In the case of the Twitter dataset, each user possesses inferred demographic information obtained from the M3 system (Wang et al., 2019b). In addition to the user's political leanings, this includes the age group (categorized as $\leq$18, 19-29, 30-39, $\geq$40), gender, and whether the Twitter account is an organization account. Additionally, the number of favorites and the account verification status were recorded. Similar to the Reddit dataset, the users' post histories were retrieved and embedded using the multilingual model `multi-qa-MiniLM-L6-cos-v1`.

**Is there a label or target associated with each instance?**
Yes, each user is assigned a label of 1 if the user has a higher frequency of posting misinformation links compared to real news links, and 0 otherwise. The determination of these links was based on domain annotations provided by Sakketou et al. (2022).

**Is any information missing from individual instances?**
No.

**Are relationships between individual instances made explicit (*e.g.*, users' movie ratings, social network links)?**
Yes. As mentioned above, each undirected edge in the Reddit dataset is created between two users if they post to the same subreddit within a 24-hour window, while each directed edge from user $A$ to user $B$ in the Twitter dataset represents user $A$ following user $B$.

**Are there recommended data splits (*e.g.*, training, development/validation, testing)?**
No, we treat the `FairGAD` problem as an unsupervised learning task, where the entire graph is utilized for training without any labeled data. In addition, it is worth noting that existing GAD methods identify anomalies based on the top nodes with high reconstruction errors, without the need for a separate classifier to be trained. Therefore, there is no need for specific data splits in this context.

**Are there any errors, sources of noise, or redundancies in the dataset?**
Yes, partially. As user demographic information is inferred for users in the Twitter dataset, we expect some noise in this category of data.

**Is the dataset self-contained, or does it link to or otherwise rely on external resources (*e.g.*, websites, tweets, other datasets)?**
Yes, our `FairGAD` datasets are self-contained.

**Does the dataset contain data that might be considered confidential (*e.g.*, data that is protected by legal privilege or by doctor–patient confidentiality, data that includes the content of individuals' non-public communications)?**
All of the raw data in the datasets are derived from publicly available sources. For Reddit, the data are directly

---

[5]`https://www.sbert.net`

accessible through the Pushshift API or Reddit dumps[6] and are also accessible through the PRAW API[7]. In the case of Twitter, the data can are directly accessible using registered Twitter API[8].

**Does the dataset contain data that, if viewed directly, might be offensive, insulting, threatening, or might otherwise cause anxiety?**
No, all of the user postings are represented as low-dimensional embeddings.

**Does the dataset identify any subpopulations (*e*.g., by age, gender)? If so, please describe how these subpopulations are identified and provide a description of their respective distributions within the dataset.**
Yes. The assignment of political leanings (*i*.e., sensitive attribute) is determined using the methodology outlined in Sakketou et al. (2022). Users are classified as right-leaning (with a sensitive label of 1) if they post a higher number of links from right-leaning sites than left-leaning sites, and as left-leaning (with a sensitive label of 0) in the opposite case. As a result, approximately 13% and 12% of the users were identified as having a positive sensitive label, indicating a right-leaning political affiliation, for the Reddit and Twitter datasets, respectively.

The assignment of misinformation spreaders (*i*.e., anomaly label) is also determined similarly to the political leanings. Users are classified as misinformation spreaders (with an anomaly label of 1) if they have a higher frequency of posting misinformation links compared to real news links, and as real news spreaders (with an anomaly label of 0) in the opposite case. As a result, approximately 14% and 7% of the users were identified as having a positive anomaly label, indicating the misinformation spreaders, for the Reddit and Twitter datasets, respectively.

Regarding other subpopulations such as age and gender, the user demographic information for the Twitter dataset was inferred using the M3 System (Wang et al., 2019b). As a result, the dataset consists of 47.7% users $\geq 40$ years old, 20.6% between 30-39 years old, 22.2% between 19-29 years old, and 9.6% $\leq 18$ years old. As for gender, it includes that 73% of users are male and 27% of users are female.

**Is it possible to identify individuals (*i*.e., one or more natural persons), either directly or indirectly (*i*.e., in combination with other data) from the dataset?**
No, it is not possible to identify individuals from the datasets as usernames are not included in the datasets.

**Does the dataset contain data that might be considered sensitive in any way (*e*.g., data that reveals race or ethnic origins, sexual orientations, religious beliefs, political opinions or union memberships, or locations; financial or health data; biometric or genetic data; forms of government identification, such as social security numbers; criminal history)?**
Yes, our `FairGAD` datasets contain data that might be considered sensitive. Based on the methodology outlined in Sakketou et al. (2022), the political affiliation and anomaly label of users were inferred from publicly posted information.

A.3 COLLECTION PROCESS

**What mechanisms or procedures were used to collect the data (*e*.g., hardware apparatuses or sensors, manual human curation, software programs, software APIs)? How were these mechanisms or procedures validated?**

---

[6]https://pushshift.io/
[7]https://praw.readthedocs.io/en/stable/index.html
[8]https://developer.twitter.com/en/products/twitter-api

We used the Pushshift API[9] to collect Reddit data, and the Twitter API[10] to collect Twitter data. All the scripts we used for data collections were written in Python.

**Who was involved in the data collection process (*e*.g., students, crowdworkers, contractors) and how were they compensated (*e*.g., how much were crowdworkers paid)?**
All data collection process was done by the authors.

**Over what timeframe was the data collected?**
The Reddit dataset encompasses the entire history of Reddit from its inception in June 2005 until March 2023. For the Twitter dataset, the user data was retrieved from Verma et al. (2022) and collected over a period of time from January 1, 2019 to July 15, 2020. The entire data collection process started on December 10, 2022 and ended on March 30, 2023.

**Were any ethical review processes conducted (*e*.g., by an institutional review board)?**
The collection of publicly available datasets was determined to be review exempt by the Institutional Review Board (IRB). Furthermore, we will strictly adhere to the privacy policies of Twitter[11] and Reddit[12] to ensure the protection of personally identifiable information when releasing our datasets.

**Did you collect the data from the individuals in question directly, or obtain it via third parties or other sources (*e*.g., websites)?**
The datasets were collected directly from the Reddit and Twitter platforms.

**Were the individuals in question notified about the data collection?**
No.

**Did the individuals in question consent to the collection and use of their data?**
No, as all the information collected is publicly available from the users' posts. The data collection process followed the privacy policies of Reddit[13] and Twitter[14].

**If consent was obtained, were the consenting individuals provided with a mechanism to revoke their consent in the future or for certain uses?**
N/A.

**Has an analysis of the potential impact of the dataset and its use on data subjects (*e*.g., a data protection impact analysis) been conducted?**
N/A.

### A.4 PREPROCESSING/CLEANING/LABELING

**Was any preprocessing/cleaning/labeling of the data done (*e*.g., discretization or bucketing, tokenization, part-of-speech tagging, SIFT feature extraction, removal of instances, processing of missing values)?**
The preprocessing and labeling process are described in Section 3. We did not perform dataset cleaning such as removal of instances.

**Was the "raw" data saved in addition to the preprocessed/cleaned/labeled data (*e*.g., to support unanticipated future uses)?**
Yes.

---

[9]https://pushshift.io/
[10]https://developer.twitter.com/en/products/twitter-api
[11]https://twitter.com/en/privacy
[12]https://www.reddit.com/policies/privacy-policy
[13]https://www.reddit.com/policies/privacy-policy
[14]https://twitter.com/en/privacy

**Is the software that was used to preprocess/clean/label the data available?**
We plan to release the code and datasets upon acceptance of the paper.

## A.5 USES

**Has the dataset been used for any tasks already?**
No, our `FairGAD` datasets were collected and curated to study the `FairGAD` problem for the first time. In this work, we evaluated the effectiveness and limitations of existing GAD methods with fairness methods in addressing the `FairGAD` problem.

**What (other) tasks could the dataset be used for?**
The datasets could be used as additional datasets for research on fair graph mining or graph anomaly detection.

**Is there anything about the composition of the dataset or the way it was collected and prepro­cessed/cleaned/labeled that might impact future uses? For example, is there anything that a future user might need to know to avoid uses that could result in unfair treatment of individuals or groups (*e*.g., stereotyping, quality of service issues) or other undesirable harms (*e*.g., financial harms, legal risks) If so, please provide a description. Is there anything a future user could do to mitigate these undesirable harms?**
The Reddit dataset was collected using the Pushshift API. Notably, Pushshift will no longer ingest new content from Reddit starting in May 2023. If prospective users wish to gather more attributes associated with our datasets, they may need to explore alternative packages like PRAW[15]. For the Twitter dataset, the information was collected using Twitter's API as of March 2023, which will require a change in the future as the API is updated for further collection.

Meanwhile, we will guarantee the availability of our datasets to the research community.

**Are there tasks for which the dataset should not be used?**
The datasets should not be used to identify political affiliation or misinformation spreaders in domains unrelated to politics or COVID-19, as the sample of users in these datasets is not representative of all users on the respective social media platforms.

## A.6 DISTRIBUTION

**Will the dataset be distributed to third parties outside of the entity (*e*.g., company, institution, organiza­tion) on behalf of which the dataset was created?**
Yes, the dataset will be freely available for distribution once accepted.

**How and When will the dataset be distributed (*e*.g., tarball on website, API, GitHub)?**
The dataset will be distributed with a publicly accessible link upon acceptance of the paper.

**Will the dataset be distributed under a copyright or other intellectual property (IP) license, and/or under applicable terms of use (ToU)?**
The dataset will be licensed under the BSD-3 Clause license.

**Have any third parties imposed IP-based or other restrictions on the data associated with the instances?**
To the best of our knowledge, no third parties have imposed IP-based or other restrictions on the dataset.

**Do any export controls or other regulatory restrictions apply to the dataset or to individual instances?**
To the best of our knowledge, there are no export controls or regulatory restrictions applicable to the dataset or individual instances.

---

[15] https://praw.readthedocs.io/en/stable/index.html

### A.7 MAINTENANCE

**How can the owner/curator/manager of the dataset be contacted (*e*.g., email address)?**
The authors can be contacted via their email or via GitHub issues.

**Is there an erratum?**
Erratas will be posted on the GitHub repository, along with version changes.

**Will the dataset be updated (*e*.g., to correct labeling errors, add new instances, delete instances')? If so, please describe how often, by whom, and how updates will be communicated to users (*e*.g., mailing list, GitHub)?**
Yes, as errors or relevant information is identified by the authors, new versions of the datasets will be made publicly available. Updates will be communicated through the GitHub repository.

**If the dataset relates to people, are there applicable limits on the retention of the data associated with the instances (*e*.g., were individuals in question told that their data would be retained for a fixed period of time and then deleted)?**
No specific limits on data retention associated with the instances are applicable.

**Will older versions of the dataset continue to be supported/hosted/maintained? If so, please describe how. If not, please describe how its obsolescence will be communicated to users.**
Yes, all versions of the datasets will be hosted and supported.

**If others want to extend/augment/build on/contribute to the dataset, is there a mechanism for them to do so? If so, please provide a description. Will these contributions be validated/verified? If so, please describe how. If not, why not? Is there a process for communicating/distributing these contributions to other users? If so, please provide a description.**
Errors and notes can be submitted via GitHub issues on the GitHub repository, where the authors will verify and discuss the submissions before updating the datasets.

## B  LIST OF POLITICS RELATED SUBREDDITS

We used a crowd-sourced collection of political subreddits[16] following previous works (Nithyanand et al., 2017).

"r/politics", "r/Liberal", "r/Conservative", "r/Anarchism", "r/LateStageCapitalism", "r/PoliticalDiscussion", "r/PoliticalHumor", "r/worldpolitics", "r/PoliticalCompassMemes", "r/PoliticalVideo", "r/PoliticalDiscourse", "r/PoliticalFactChecking", "r/PoliticalRevisionism", "r/PoliticalIdeology", "r/PoliticalRevolution", "r/PoliticalMemes", "r/PoliticalModeration", "r/PoliticalCorrectness", "r/PoliticalCorrectnessGoneMad", "r/PoliticalTheory", "r/PoliticalQuestions", "r/PoliticalScience", "r/PoliticalHumorModerated", "r/PoliticalCompass", "r/PoliticalDiscussionModerated", "r/worldnews", "r/news", "r/worldpolitics", "r/worldevents", "r/business", "r/economics", "r/environment", "r/energy", "r/law", "r/education", "r/history", "r/PoliticsPDFs", "r/WikiLeaks", "r/SOPA", "r/NewsPorn", "r/worldnews2", "r/AnarchistNews", "r/republicofpolitics", "r/LGBTnews", "r/politics2", "r/economic2", "r/environment2", "r/uspolitics", "r/AmericanPolitics", "r/AmericanGovernment", "r/ukpolitics", "r/canada", "r/euro", "r/Palestine", "r/eupolitics", "r/MiddleEastNews", "r/Israel", "r/india", "r/pakistan", "r/china", "r/taiwan", "r/iran", "r/russia", "r/Libertarian", "r/Anarchism", "r/socialism", "r/progressive", "r/Conservative", "r/americanpirateparty", "r/democrats", "r/Liberal", "r/new_right", "r/Republican", "r/egalitarian", "r/demsocialist", "r/LibertarianLeft", "r/Liberty", "r/Anarcho_Capitalism", "r/alltheleft", "r/neoprogs",

---

[16]https://www.reddit.com/r/redditlists/comments/josdr/list_of_political_subreddits/

"r/democracy", "r/peoplesparty", "r/Capitalism", "r/Anarchist", "r/feminisms", "r/republicans", "r/Egalitarianism", "r/anarchafeminism", "r/Communist", "r/socialdemocracy", "r/conservatives", "r/Freethought", "r/StateOfTheUnion", "r/equality", "r/propagandaposters", "r/SocialScience", "r/racism", "r/corruption", "r/propaganda", "r/lgbt", "r/feminism", "r/censorship", "r/obama", "r/war", "r/antiwar", "r/climateskeptics", "r/conspiracyhub", "r/infograffiti", "r/CalPolitics", "r/politics_new"

## C    FURTHER GRAPH PROPERTIES

**Details on the Inherent Biases of the Graph.** In this section, we describe the equations for the attribute and structural biases (Dong et al., 2022), which are used to analyze our `FairGAD` datasets (see Section 3).

*Attribute Bias.* Let $\mathbf{X}_{norm} \in \mathbb{R}^{N \times M}$ denote a normalized attribute matrix of an input graph, where $N$ and $M$ represent the numbers of nodes and attributes, respectively. Given $\mathbf{X}_{norm}$, the attribute bias $b_{attr}$ is calculated as follows (Dong et al., 2022):

$$b_{attr} = \frac{1}{M} \sum_{m=1}^{M} W(pdf(\mathcal{X}_m^0), pdf(\mathcal{X}_m^1)), \tag{1}$$

where $\mathcal{X}_m^0$ and $\mathcal{X}_m^1$ denote the $m$-th attribute value sets for nodes with sensitive attributes of 0 and 1, respectively. That is, we divide the attributes of all nodes as $\{(\mathcal{X}_1^0, \mathcal{X}_1^1), (\mathcal{X}_2^0, \mathcal{X}_2^1), \cdots, (\mathcal{X}_M^0, \mathcal{X}_M^1)\}$. Also, $W$ and $pdf$ denote the Wasserstein-1 distance (Villani, 2021) between two distributions and the probability density function for a set of values, respectively.

*Structural Bias.* We denote a normalized adjacency matrix with re-weighted self-loops as $\mathbf{P}_{norm} = \alpha \mathbf{A}_{norm} + (1 - \alpha)\mathbf{I}$, where $\mathbf{A}_{norm}$ and $\mathbf{I}$ represent the symmetric normalized adjacency matrix and the identity matrix, respectively; $\alpha$ is a hyperparameter ranging from 0 to 1. Then, the propagation matrix is defined as $\mathbf{M}_H = \sum_{h=1}^{H} \beta^h \mathbf{P}_{norm}^h$, where $H$ and $\beta$ indicate the number of hops to measure for the propagation and the discount factor that reduces the weight of propagation from the neighbors with higher hops, respectively. Given $\mathbf{M}_H$, the structural bias $b_{struc}$ is calculated as follows (Dong et al., 2022):

$$b_{struc} = \frac{1}{M} \sum_{m=1}^{M} W(pdf(\mathcal{R}_m^0), pdf(\mathcal{R}_m^1)), \tag{2}$$

where $\mathbf{R} = \mathbf{M}_H \mathbf{X}_{norm}$ represents the reachability matrix. That is, $\mathcal{R}_m^0$ and $\mathcal{R}_m^1$ represent the $m$-th attribute value sets in $\mathbf{R}$ for nodes with sensitive attributes of 0 and 1, respectively.

In addition, we analyze the potential for structural bias based on changes in the $k$-hop neighborhood information (from EDITS) in our datasets (*i.e.*, Reddit and Twitter), as well as in existing datasets (*i.e.*, German, Credit, and Bail). Figure I shows the results. In general, bias metrics exhibit a monotonic increase for hops of two or more, potentially due to nodes with similar sensitive attributes being more strongly connected. Consequently, we suggest adhering to the metrics outlined in EDITS for a more accurate characterization of the dataset.

**Additional Properties of the Graph.** In Table I, we provide the additional properties of three synthetic datasets (*i.e.*, German, Credit, and Bail) and our `FairGAD` datasets (*i.e.*, Reddit and Twitter). Following the FairGen paper (Zheng et al., 2023), we used the following four properties:

- Triangles: the number of three mutually connected nodes in the graph, *i.e.*, $\frac{|\{u,v,w\}|(u,v)(v,w)(u,w)\subseteq\mathcal{E}|}{6}$, where $\mathcal{E}$ indicates the set of edges.

- Exponent in Power-law: the exponent of the power-law distribution, *i.e.*, $1 + n(\sum_{u \in \mathcal{V}} \log(\frac{d(u)}{d_{min}}))^{-1}$, where $n$, $d(u)$, and $d_{min}$ indicate the number of nodes, the degree of node $u$, and the minimum degree in the graph, respectively.

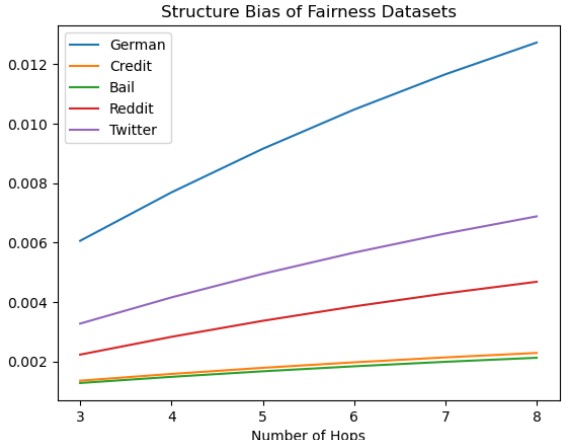

Figure I: Values of Structural Bias with varying $k$-hop neighborhoods

- Gini: the Gini coefficient of the degree distribution, $i.e.,$ $\frac{2 \sum_{u \in \mathcal{V}} u d(u)}{n \sum_{u \in \mathcal{V}} d(u)} - \frac{n+1}{n}$

- Entropy: the entropy of the degree distribution, $i.e.,$ $\frac{1}{\log n} \sum_{u \in \mathcal{V}} -\frac{d(u)}{|\mathcal{V}|} \log \frac{d(u)}{|\mathcal{V}|}$

Table I: Additional graph properties of our datasets

|  | German | Credit | Bail | Reddit | Twitter |
|---|---|---|---|---|---|
| **Triangles** | 116,604 | 58,794,906 | 2,014,022 | 24,041,458 | 743,915 |
| **Exponent in Power-law** | 2.162 | 1.604 | 1.799 | 1.577 | 2.194 |
| **Gini** | 0.287 | 0.493 | 0.324 | 0.610 | 0.708 |
| **Entropy** | 1.759 | 1.788 | 1.818 | 1.699 | 1.648 |

## D    FURTHER DETAILS ON FAIRNESS REGULARIZERS

**FairOD** (Shekhar et al., 2021) is a fairness method that originated in unsupervised anomaly detection. The method focuses on improving fairness through EOO, stating how improving SP alone can lead to models lazily predicting an equal number of anomalies for each sensitive attribute. Thus, it introduces two losses $\mathcal{L}_{FairOD}$ and $\mathcal{L}_{ADCG}$. First, the SP loss $\mathcal{L}_{FairOD}$ is defined as follows (Shekhar et al., 2021):

$$\mathcal{L}_{FairOD} = \left| \left(1 - \frac{1}{n}\right)^2 \frac{\left(\sum_{i=1}^{n} Err(v_i)\right) \left(\sum_{i=1}^{n} S(v_i)\right)}{\sigma_{Err} \sigma_S} \right|, \tag{3}$$

where $Err(v_i)$ and $S(v_i)$ represent the reconstruction error and sensitive attribute of node $v_i$, respectively. Also, $\sigma_{Err}$ and $\sigma_S$ represent the standard deviations of the reconstruction error and sensitive attribute, respectively, across all nodes. This loss is used to reduce SP.

An additional loss $\mathcal{L}_{ADCG}$ is used to reduce EOO, which ensures that the fair model has a similar ranking of anomaly nodes with an original model without the bias term. It uses an approximation of the discounted

cumulative gain ("ADCG") that is differentiable with the following formula (Shekhar et al., 2021):

$$\mathcal{L}_{ADCG} = \sum_{s \in \{0,1\}} \left( 1 - \sum_{\{v_i : S(v_i) = s\}} \frac{2^{BaseErr(v_i)} - 1}{\log_2\left(1 + IDCG_{S=s} \cdot DIFF(v_i)\right)} \right), \tag{4}$$

where $BaseErr(v_i)$ indicates the reconstruction error of node $v_i$ in the original model. In addition, $DIFF(v_i) = \sum_{\{v_j : S(v_j) = s\}} sigmoid(Err(v_j) - Err(v_i))$ is the differentiable ranking loss utilizing the sigmoid function, and $IDCG_{S=s} = \sum_{j=1}^{|\{v_j : S(v_j) = s\}|} (2^{BaseErr(v_j)} - 1)/(\log_2(1+j))$ is the ideal discounted cumulative gain, *i.e.*, the greatest possible value across all nodes in each sensitive attribute group. Thus, the two losses encourage the model to fairly predict anomalies across the sensitive attribute groups, while preserving the original ranking of anomalies as much as possible to reduce the impact on performance.

The **correlation** regularizer is another loss defined in the FairOD implementation[17]. This loss measures the correlation between sensitive attributes and node representation errors by using the cosine rule. This ensures that nodes are encoded to achieve similar accuracy, regardless of any sensitive attributes. Unlike FairOD, this loss does not directly consider a fairness metric in its formula. The loss $\mathcal{L}_{corr}$ is calculated as follows:

$$\mathcal{L}_{corr} = \left| \frac{(\mathbf{Err} \cdot \mathbf{S})}{\sqrt{(\mathbf{Err} \cdot \mathbf{Err})(\mathbf{S} \cdot \mathbf{S})}} \right|, \tag{5}$$

where $\mathbf{Err}$ represents the vector of reconstruction errors across all nodes, and $(\mathbf{X} \cdot \mathbf{Y})$ represents the dot product of two vectors $\mathbf{X}$ and $\mathbf{Y}$.

**HIN** (Zeng et al., 2021) is another regularizer for fairness representation learning. While it originally intends for heterogeneous information networks, the loss function can be adapted to GAD to reduce the same SP fairness metric. The loss $\mathcal{L}_{HIN}$ penalizes the difference in prediction rates between sensitive attribute groups for both anomalies and non-anomalies. The loss $\mathcal{L}_{HIN}$ is calculated as follows (Zeng et al., 2021):

$$\mathcal{L}_{HIN} = \sum_{y \in \{0,1\}} \left( \frac{\sum_{\{v : S(v) = 1\}} Pr(\hat{y}_v = y)}{|\{v : S(v) = 1\}|} - \frac{\sum_{\{v : S(v) = 0\}} Pr(\hat{y}_v = y)}{|\{v : S(v) = 0\}|} \right)^2, \tag{6}$$

where $Pr(\hat{y}_v = 1)$ indicates the probability that node $v$ is predicted as an anomaly. Note that the HIN regularizer introduces another function that reduces EOO, but requires labels. Thus, as mentioned in Section 4.1, we used $\mathcal{L}_{ADCG}$ from the FAIROD regularizer as a replacement.

## E    FURTHER RESULTS ON FAIRNESS REGULARIZERS

We present the omitted results for the trade-off spaces of COLA in Figure II.

Additionally, we present the AUCROC and EOO results for each of the GAD methods, *i.e.*, COLA, CONAD, DOMINANT, with the fairness regularizers, *i.e.*, FAIROD and HIN, on our `FairGAD` datasets, *i.e.*, Reddit and Twitter (see Figures III–XII). Note that the results of GAD methods with CORRELATION loss on the Reddit dataset are already presented in Section 4.2. Figure XIII shows the results on the Twitter dataset.

## F    PRELIMINARY RESULTS ON COMBINATION OF GRAPH DEBIASERS AND FAIRNESS REGULARIZERS

In this section, we present our preliminary findings on combining graph debiasers (*i.e.*, FAIRWALK and EDITS) with fairness regularizers (*i.e.*, FAIROD, CORRELATION, and HIN). This is naïvely done by treating

---

[17]https://github.com/Shubhranshu-Shekhar/fairOD

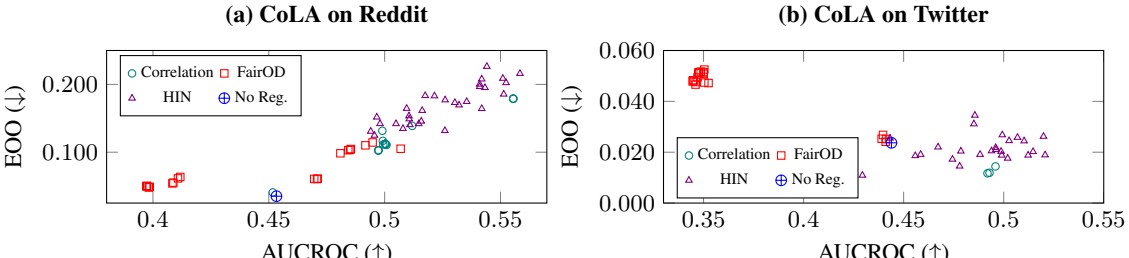

Figure II: Trade-off spaces for CoLA with fairness regularizers.

the debiased graph via FAIRWALK or EDITS as a new dataset and simply applying the GAD methods with fairness regularizers on the debiased graph. The trade-off spaces are shown in Figures XIV–XVI.

We believe that there are more sophisticated approaches for their integration: (1) the attribute bias optimization in EDITS could be modified to emphasize the sensitive attribute through the formulations of the regularizers, rather than relying on the difference in distributions; (2) graph debiasers could be more tightly integrated with GAD methods; and (3) another example would be combining FAIRWALK and COLA, ensuring that the neighborhood sampling performed by COLA for generating instance pairs is conducted in a fair manner similar to FAIRWALK. These approaches suggest potential avenues for further research and development in the synergistic combination of graph debiasers and fairness regularizers.

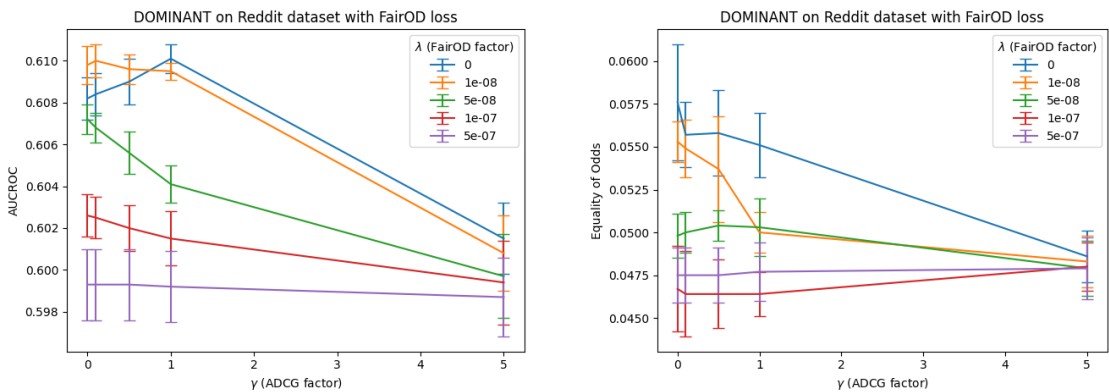

Figure III: Accuracy (AUCROC, left, higher is better) and Fairness (Equality of Odds, right, lower is better) metric for DOMINANT with the FairOD regularizer on our Reddit dataset.

## G  FURTHER RESULTS ON ALTERNATIVE SENSITIVE ATTRIBUTES

We conducted some preliminary experiments that consider the age and gender of users, as inferred by the M3 system, as alternative sensitive attributes. It should be noted that (1) the inference of age and gender was necessitated by the lack of direct demographic information from platforms like Twitter and Reddit, (2) the experiments were exclusively performed on the Twitter dataset due to the incompatibility of the Reddit dataset with the requirements of the M3 system (*i.e.*, general lack of profile images, biographies, and names for Reddit accounts), and (3) there are no reputable demographic inference systems for the Reddit dataset.

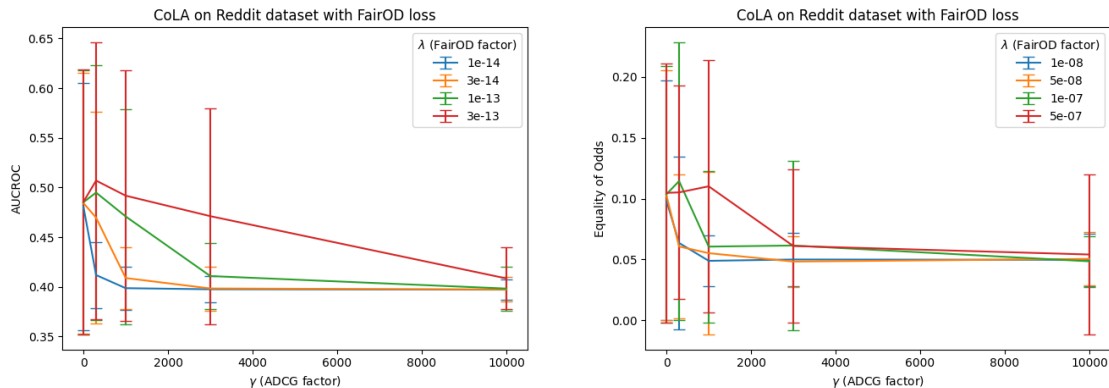

Figure IV: Accuracy (AUCROC, left, higher is better) and Fairness (Equality of Odds, right, lower is better) metric for CoLA with the FairOD regularizer on our Reddit dataset.

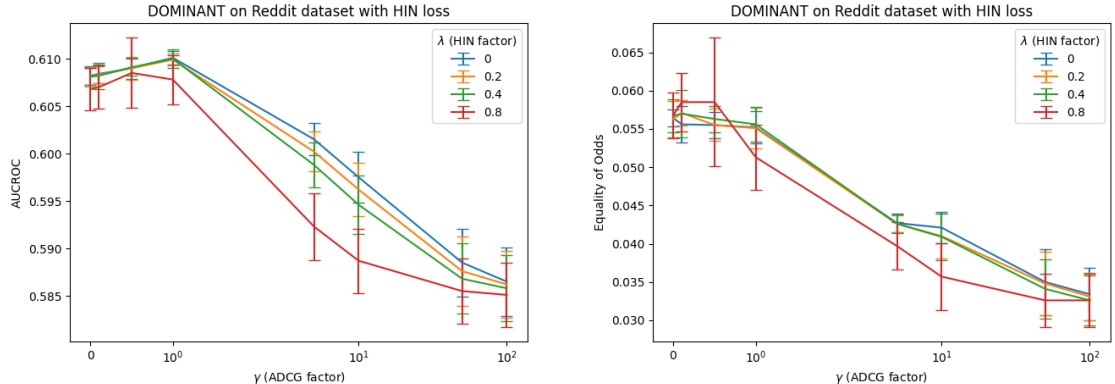

Figure V: Accuracy (AUCROC, left, higher is better) and Fairness (Equality of Odds, right, lower is better) metric for DOMINANT with the HIN regularizer on our Reddit dataset.

Specifically, we created two new versions of our datasets, one in which the sensitive attribute is gender (the M3 system infers that the user would be a female with a greater than 0.5 chance), and age (the M3 system infers that the user is more probable to be in the category of ">=40 years old" than "<=18", "19-29", "30-39" categories). We first run the original GAD methods CoLA, CONAD, and DOMINANT without fairness methods on these datasets to analyze changes in accuracy and fairness metrics.

Table II shows the different values of the fairness metric (*i.e.*, Equality of Odds (EOO), lower is better) as the sensitive attribute changes. Note that the accuracy, as measured by AUCROC, remains relatively constant because the node attributes and the network structure remain unchanged throughout the shifts in the sensitive attribute. On the other hand, fairness, as measured by EOO in this table, shows the tendency for different levels of unfairness to manifest across different sensitive attributes. By choosing the user's political leanings as the sensitive attribute, a Twitter dataset with increased levels of unfairness was created. Thus, we believe it would be easier to analyze the difference in fairness metrics after applying fairness regularizers or graph debiasers.

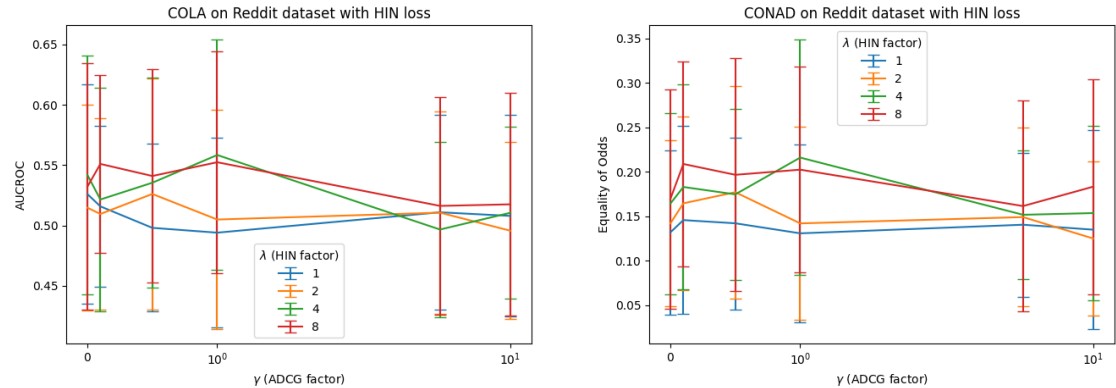

Figure VI: Accuracy (AUCROC, left, higher is better) and Fairness (Equality of Odds, right, lower is better) metric for CoLA with the HIN regularizer on our Reddit dataset.

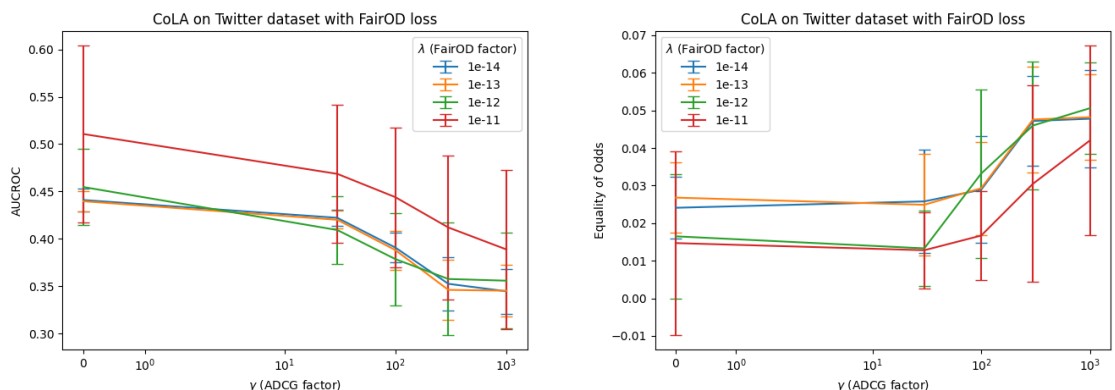

Figure VII: Accuracy (AUCROC, left, higher is better) and Fairness (Equality of Odds, right, lower is better) metric for CoLA with the FairOD regularizer on our Twitter dataset.

Table II: Results for the fairness metric (Equality of Odds, lower is better) across different sensitive attributes for the Twitter dataset using the original GAD methods CoLA, CONAD, and DOMINANT

| Twitter (Equality of Odds) | | | |
| --- | --- | --- | --- |
| Sensitive Attribute | Political Leaning | Gender | Age |
| CoLA | 0.023±0.012 | 0.007±0.004 | 0.017±0.006 |
| CONAD | 0.044±0.003 | 0.030±0.004 | 0.036±0.007 |
| DOMINANT | 0.044±0.003 | 0.031±0.004 | 0.037±0.007 |

Figures XVII, XVIII, and XIX demonstrate how the fairness regularizers are affected by changing the sensitive attribute. We only conducted experiments on the Twitter dataset where gender was used as the sensitive attribute for the DOMINANT method. We can see that the fairness regularizers exhibit similar trends when applied to this new setting. Generally, increasing $\lambda$ leads to lower accuracy (*i.e.*, AUCROC) and unfairness

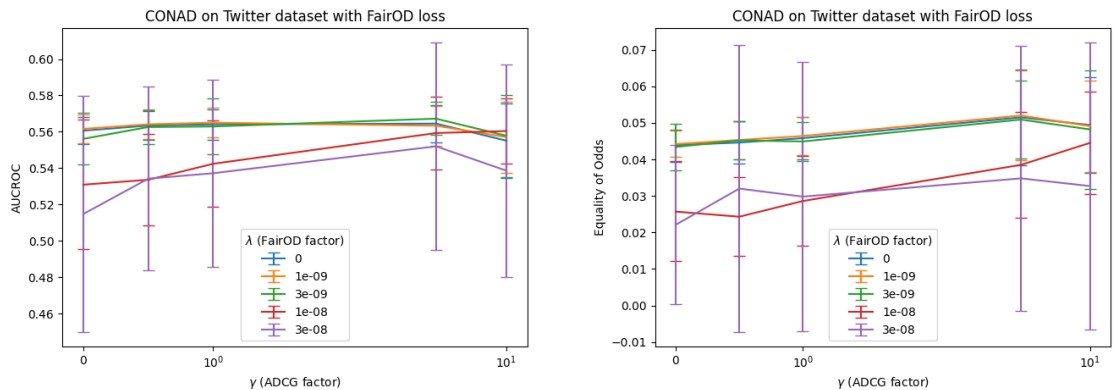

Figure VIII: Accuracy (AUCROC, left, higher is better) and Fairness (Equality of Odds, right, lower is better) metric for CONAD with the FairOD regularizer on our Twitter dataset.

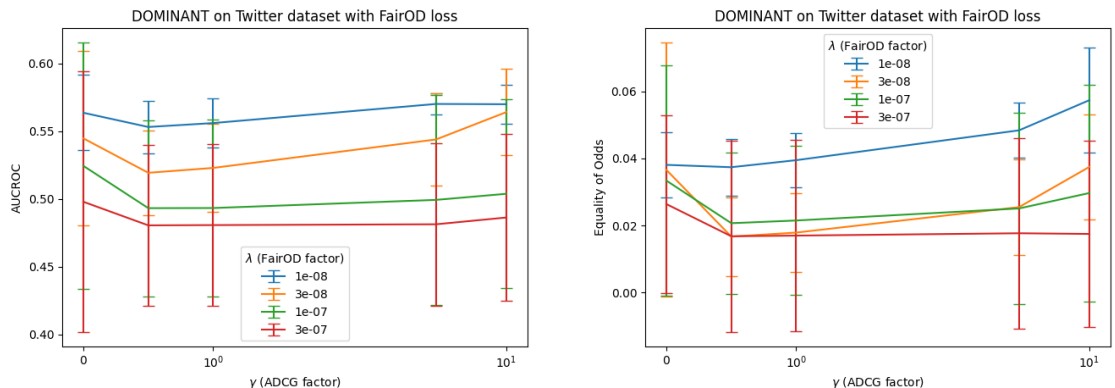

Figure IX: Accuracy (AUCROC, left, higher is better) and Fairness (Equality of Odds, right, lower is better) metric for DOMINANT with the FairOD regularizer on our Twitter dataset.

(*i.e.*, EOO), which is expected for a regularizer attempting to enhance fairness. Increasing $\gamma$ can also boost accuracy, but only to a certain point beyond which the regularizer overpenalizes the model, causing it to fail to match the noise in the base model's predictions. This demonstrates that our fairness regularizers are applicable to similar datasets with different sensitive attributes.

## H   HUMAN VERIFICATION FOR THE CLASSIFICATION OF POLITICAL LEANING LABELS

We randomly sampled 1,000 posts in the Reedit dataset that contain URLs with either a left or right political leaning. For each politics-related URL, we asked 3 annotators who have no conflict of interest with the authors to assess the users' political leanings as expressed by their sharing behavior. To achieve this, the annotators carefully review the content associated with the URL as well as the corresponding post. Our

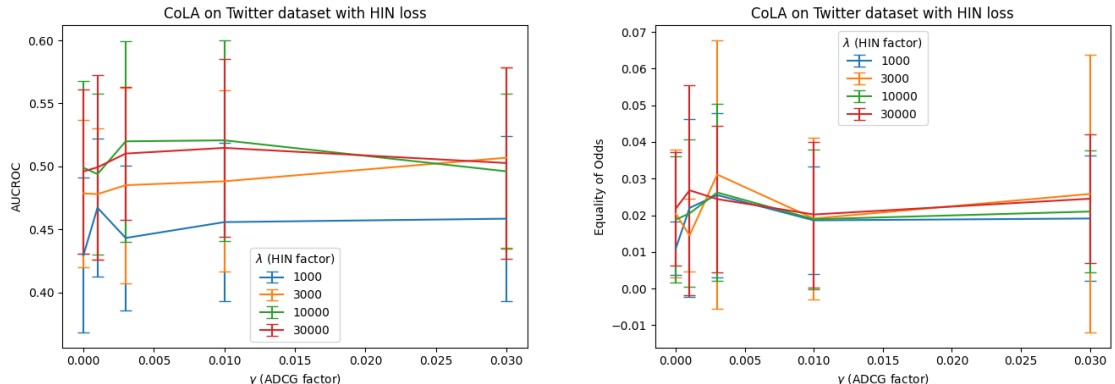

Figure X: Accuracy (AUCROC, left, higher is better) and Fairness (Equality of Odds, right, lower is better) metric for CoLA with the HIN regularizer on our Twitter dataset.

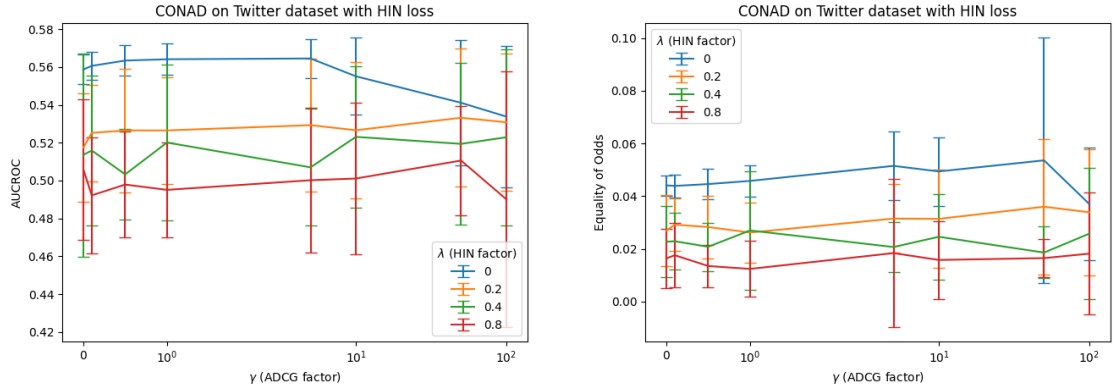

Figure XI: Accuracy (AUCROC, left, higher is better) and Fairness (Equality of Odds, right, lower is better) metric for CONAD with the FairOD regularizer on our Twitter dataset.

guiding principle is that if the user expresses support for the content in the post, we consider the user to be aligned with the political leanings of that post, and vice versa.

The annotators agree upon 99.8% of the examples with a Fleiss' Kappa of 0.793. Here, Fleiss' Kappa is a statistical measure used to determine the reliability of agreement between multiple raters. Fleiss' Kappa ranges from -1 to 1, with a value less than 0.2 indicating low agreement, a value in the range [0.4, 0.6] indicating moderate agreement, and a value greater than 0.6 indicating significant agreement. After applying a majority vote across the ratings of the 3 annotators, we found that in 99.6% of the cases, the user of the post supports the political leaning associated with the URL in the post. In summary, we empirically demonstrated that website hyperlinks shared by users can be used to classify the political leanings of users.

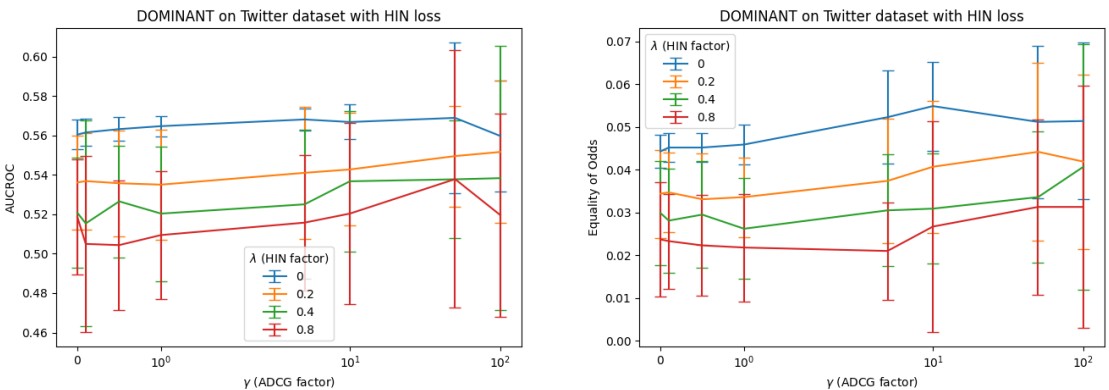

Figure XII: Accuracy (AUCROC, left, higher is better) and Fairness (Equality of Odds, right, lower is better) metric for DOMINANT with the FairOD regularizer on our Twitter dataset.

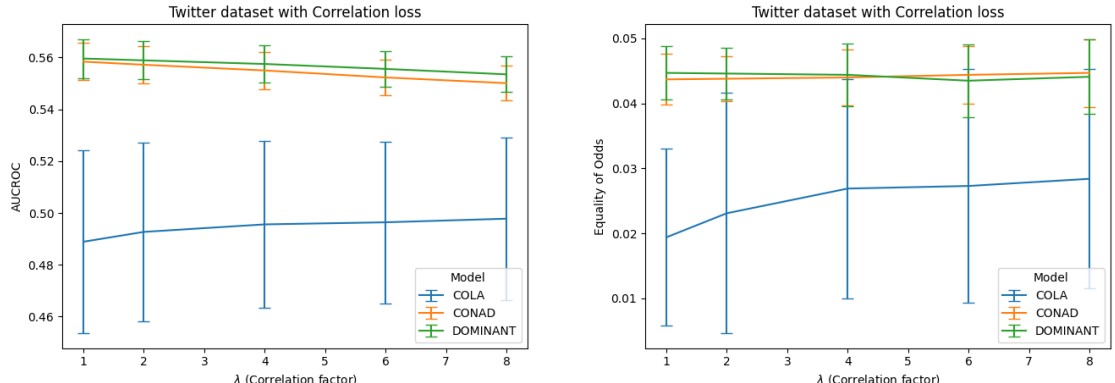

Figure XIII: Accuracy (AUCROC, left, higher is better) and Fairness (Equality of Odds, right, lower is better) metric for all GAD models with the Correlation regularizer on our Twitter dataset.

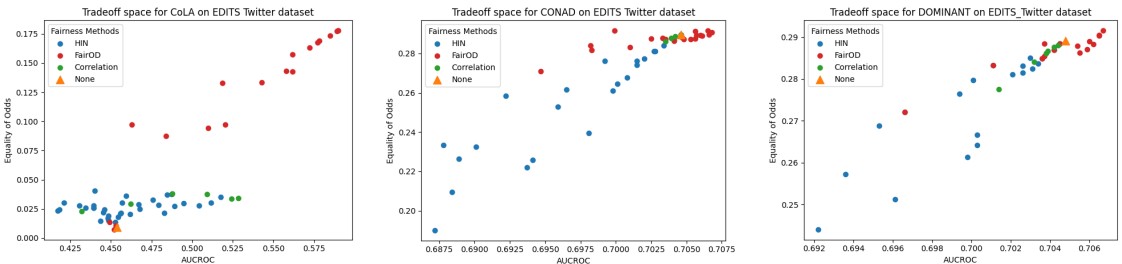

Figure XIV: Accuracy-fairness trade-off space for the debiased Twitter dataset via EDITS.

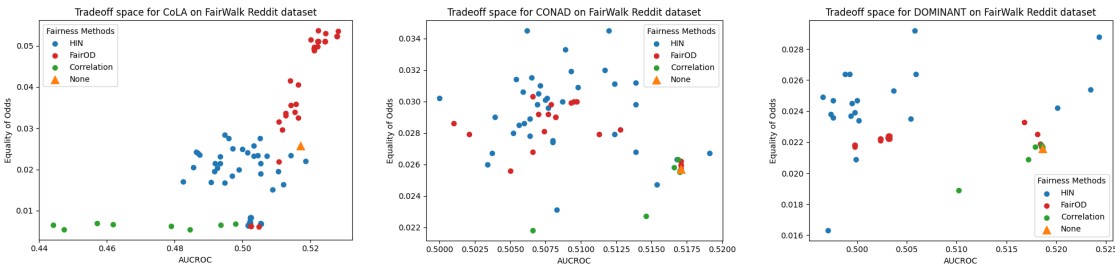

Figure XV: Accuracy-fairness trade-off space for the debiased Reddit dataset via FAIRWALK.

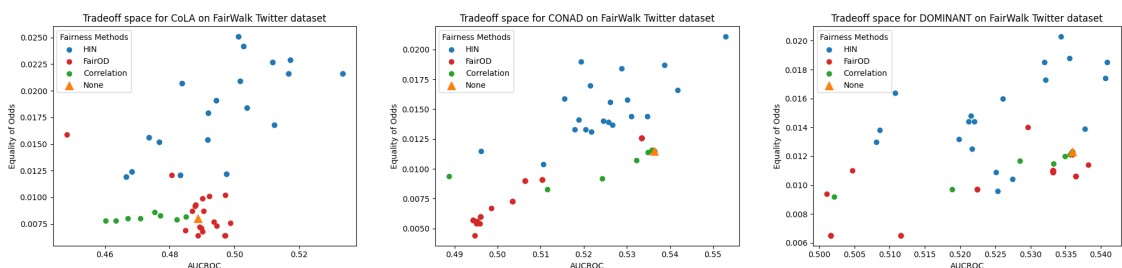

Figure XVI: Accuracy-fairness trade-off space for the debiased Twitter dataset via FAIRWALK.

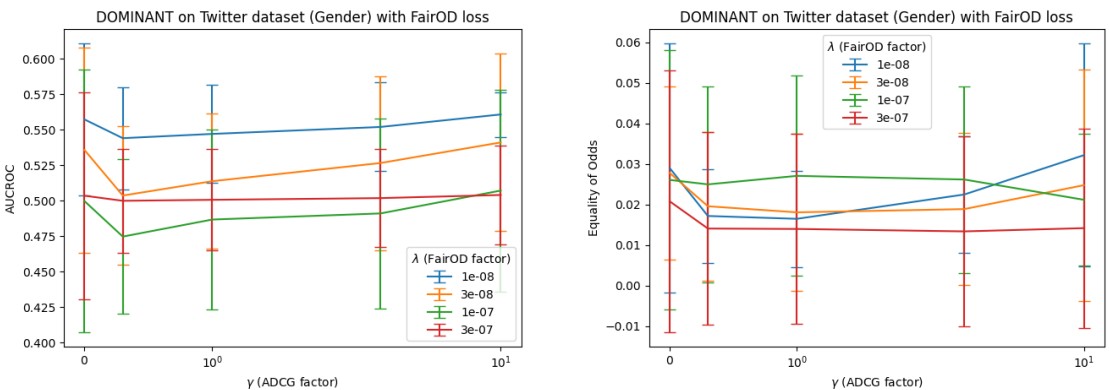

Figure XVII: Accuracy (AUCROC, left, higher is better) and Fairness (Equality of Odds, right, lower is better) metric for DOMINANT with the FairOD regularizer on the Twitter dataset with gender as a sensitive attribute.

## I   FURTHER RESULTS ON ADDITIONAL GAD AND AD METHODS

We conducted experiments with a recent GAD method (*i.e.*, VGOD (Huang et al., 2023)), five non-GNN-based anomaly detection methods (*i.e.*, DONE (Bandyopadhyay et al., 2020), AdONE (Bandyopadhyay et al., 2020), ECOD (Li et al., 2022), VAE (Kingma & Welling, 2014), and ONE (Bandyopadhyay et al., 2019)), and two heuristic methods (*i.e.*, LOF (Breunig et al., 2000) and IF (Liu et al., 2008)) on our datasets in terms

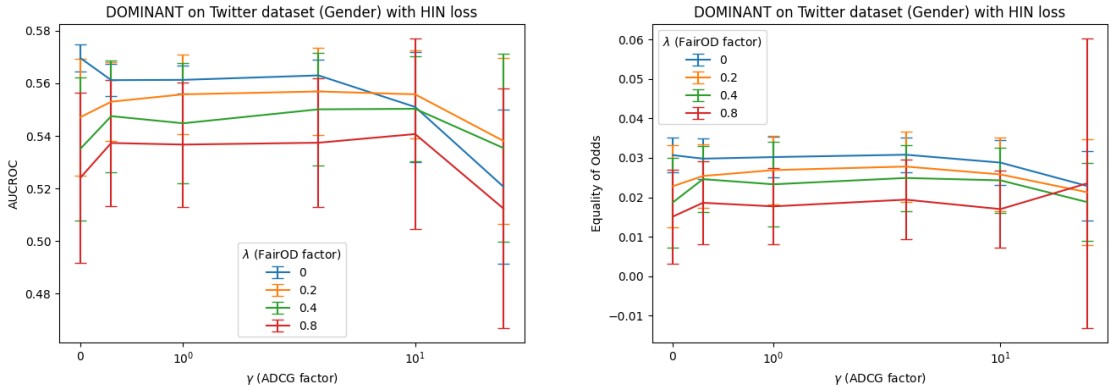

Figure XVIII: Accuracy (AUCROC, left, higher is better) and Fairness (Equality of Odds, right, lower is better) metric for DOMINANT with the HIN regularizer on the Twitter dataset with gender as a sensitive attribute.

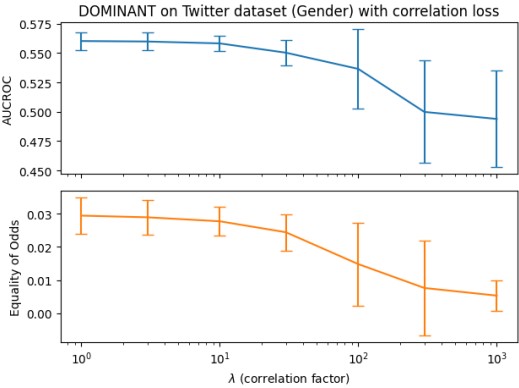

Figure XIX: Accuracy (AUCROC, top, higher is better) and Fairness (Equality of Odds, bottom, lower is better) metric for DOMINANT with the correlation regularizer on the Twitter dataset with gender as a sensitive attribute.

of accuracy and fairness. We ran experiments to show their effects when applied to our datasets, together with graph debiasers, in Table III and Table IV.

In summary, all of the new baselines show an intermediary position in terms of accuracy, positioned between the CoLA method and the GNN-based methods. For example, it is worth noting that due to their individual treatment of attributes, DONE and AdONE are sensitive to attribute bias in the datasets. The EOO of these methods is larger than other methods on the Reddit dataset, whereas on the Twitter dataset, it is smaller than other methods. This divergence is a reflection of the more pronounced attribute bias in the Reddit dataset. Furthermore, the accuracy of these methods is also higher on the debiased graphs from FAIRWALK, given that the attributes in these graphs also encompass the graph structure, which aids DONE and AdONE in their analysis. Nonetheless, this also results in a higher EOO (more unfairness). This phenomenon arises due to the propagation of structural bias into node attributes, thereby reinforcing the unfairness inherent in the graph structure. In addition, ONE is not effective in this problem due to the challenge of capturing nonlinear

Table III: Accuracy (AUCROC, higher is better) and Fairness (EOO, lower is better) metrics across the different anomaly detection methods on our Twitter dataset. Some missing results will be added before publication of our paper.

| Twitter | | | | | | |
|---|---|---|---|---|---|---|
| Debiaser | None | | EDITS | | FairWalk | |
| Metric | AUCROC | EOO | AUCROC | EOO | AUCROC | EOO |
| CoLA | 0.443±0.006 | 0.023±0.012 | 0.452±0.013 | 0.009±0.005 | 0.488±0.006 | 0.001±0.001 |
| CONAD | 0.558±0.007 | 0.044±0.003 | 0.704±0.001 | 0.278±0.004 | 0.536±0.009 | 0.013±0.002 |
| DOMINANT | 0.560±0.007 | 0.044±0.003 | 0.704±0.001 | 0.278±0.003 | 0.535±0.009 | 0.013±0.002 |
| VGOD | 0.736±0.006 | 0.111±0.021 | 0.823±0.032 | 0.144±0.085 | 0.602±0.003 | 0.052±0.004 |
| DONE | 0.507±0.023 | 0.025±0.015 | 0.577±0.031 | 0.088±0.028 | 0.590±0.014 | 0.079±0.012 |
| AdONE | 0.522±0.026 | 0.023±0.010 | 0.578±0.032 | 0.101±0.033 | 0.594±0.014 | 0.085±0.013 |
| ECOD | 0.454±0.000 | 0.018±0.000 | 0.454±0.000 | 0.018±0.000 | 0.704±0.000 | 0.157±0.000 |
| VAE | 0.456±0.000 | 0.019±0.000 | 0.457±0.000 | 0.019±0.000 | 0.708±0.000 | 0.158±0.000 |
| ONE | 0.501±0.005 | 0.010±0.008 | 0.501±0.005 | 0.010±0.008 | 0.544±0.005 | 0.025±0.011 |
| LoF | 0.460±0.000 | 0.029±0.000 | 0.451±0.000 | 0.035±0.000 | 0.500±0.000 | 0.010±0.000 |
| IF | 0.461±0.003 | 0.015±0.005 | 0.461±0.010 | 0.018±0.001 | 0.699±0.002 | 0.145±0.014 |

Table IV: Accuracy (AUCROC, higher is better) and Fairness (EOO, lower is better) metrics across the different anomaly detection methods on our Reddit dataset. Runs are marked 'o.o.m.' if they encountered an out-of-GPU memory error. Some missing results will be added before publication of our paper.

| Reddit | | | | | | |
|---|---|---|---|---|---|---|
| Debiaser | None | | EDITS | | FairWalk | |
| Metric | AUCROC | EOO | AUCROC | EOO | AUCROC | EOO |
| CoLA | 0.453±0.014 | 0.177±0.014 | o.o.m | o.o.m | 0.502±0.004 | 0.003±0.003 |
| CONAD | 0.608±0.001 | 0.055±0.002 | o.o.m | o.o.m | 0.517±0.024 | 0.028±0.018 |
| DOMINANT | 0.608±0.001 | 0.057±0.003 | o.o.m | o.o.m | 0.518±0.023 | 0.025±0.017 |
| VGOD | 0.721±0.009 | 0.472±0.063 | o.o.m | o.o.m | 0.673±0.002 | 0.295±0.006 |
| DONE | 0.578±0.033 | 0.068±0.043 | o.o.m | o.o.m | 0.600±0.011 | 0.148±0.015 |
| AdONE | 0.575±0.027 | 0.077±0.048 | o.o.m | o.o.m | 0.607±0.011 | 0.157±0.015 |
| ECOD | 0.578±0.000 | 0.098±0.000 | o.o.m | o.o.m | 0.736±0.000 | 0.467±0.000 |
| VAE | 0.580±0.000 | 0.098±0.000 | o.o.m | o.o.m | 0.735±0.000 | 0.474±0.000 |
| ONE | 0.496±0.007 | 0.014±0.009 | o.o.m | o.o.m | 0.524±0.008 | 0.035±0.021 |
| LoF | 0.597±0.000 | 0.088±0.000 | o.o.m | o.o.m | 0.614±0.000 | 0.162±0.000 |
| IF | 0.580±0.003 | 0.095±0.007 | o.o.m | o.o.m | 0.725±0.008 | 0.428±0.019 |

relationships using the matrix factorization techniques. Additionally, ONE assigns equal weight to all node attributes, which may not be necessary given the large number of attributes for each node.

On the other hand, VGOD performs very well in both datasets. The main contribution of VGOD stems from its capacity to capture neighborhood variance as a determining factor in identifying structural outliers. This is achieved by sampling both positive and negative edge sets, while attribute outliers are handled in a standard manner comparable to other GAD methods by reconstructing the node attributes. As such, the improved

accuracy (AUCROC) can be attributed to the detection of more structural outliers. However, this also leads to an increase in unfairness (EOO), which is not desirable in FairGAD. We posit that this is because of the significant discriminatory effect of VGOD in identifying neighborhood variances. As a result, structural outliers of nodes with the majority sensitive attribute are more likely to be identified since they form a larger proportion of both positive and negative edges.

We conducted further experiments on the fairness regularizers with the AdONE method on the Reddit dataset, as shown in Figure XX, Figure XXI and Figure XXII. Our results show that the fairness regularizers introduced in our analysis can be extended to other methods as well. However, the effects of increasing $\lambda$ and $\gamma$ may vary depending on the model. In particular, since AdONE is an adversarial model, it would work differently compared to the other GAD methods analyzed, which rely on self-supervised learning or reconstruction error. This is also evidenced by the large error bars for each run due to the less stable nature of adversarial learning. However, we see that when using the HIN regularizer, increasing $\lambda$ can actually lead to an increase in model performance, unlike other regularizers, but also leads to a wider range in EOO (*i.e.*, unfairness). More analysis is needed to understand how the fairness regularizers work with AdONE and DONE.

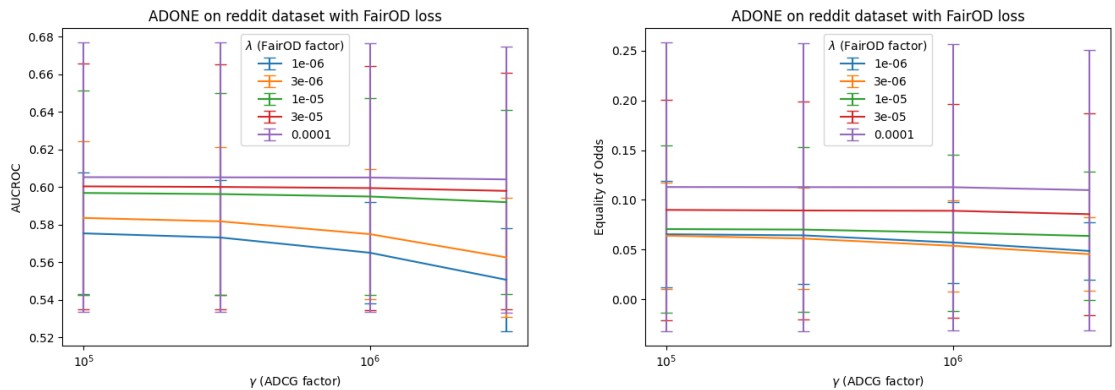

Figure XX: Accuracy (AUCROC, left, higher is better) and Fairness (Equality of Odds, right, lower is better) metric for AdONE with the FairOD regularizer on the Reddit dataset.

Lastly, we conducted further investigation of the trade-off space for VGOD on our Reddit and Twitter datasets as shown in Figure XXIII. The findings indicate the range of trade-offs for VGOD across our datasets from both Reddit and Twitter. It is important to note that the standard deviation for each result is considerably high, making the trade-off curve more of a rough estimation. The trade-off curve for the Reddit dataset appears linear, resembling the curves for DOMINANT and CONAD, which reflects the challenge of balancing fairness and accuracy in a dataset that is inherently more unfair, such as Reddit. For the Twitter dataset, it should be noted that there is high variance in AUCROC within a narrow range. Fairness regularizers appear to have some effect on reducing unfairness while maintaining accuracy; however, the impact is not significant enough to draw a definitive conclusion. Therefore, further research is needed to identify a GAD method that prioritizes fairness.

## J  FURTHER RESULTS FOR REMOVING SENSITIVE ATTRIBUTES

We conducted experiments that remove the sensitive attribute on the Reddit dataset. Table V shows the results.

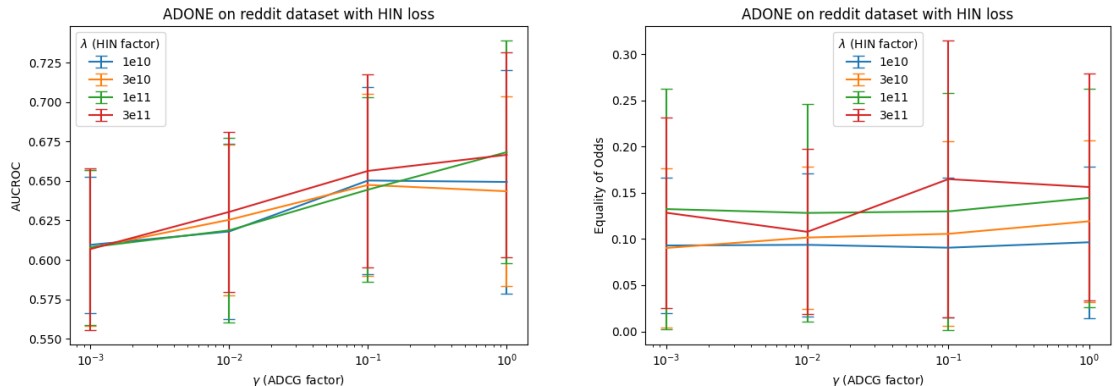

Figure XXI: Accuracy (AUCROC, left, higher is better) and Fairness (Equality of Odds, right, lower is better) metric for AdONE with the HIN regularizer on the Reddit dataset.

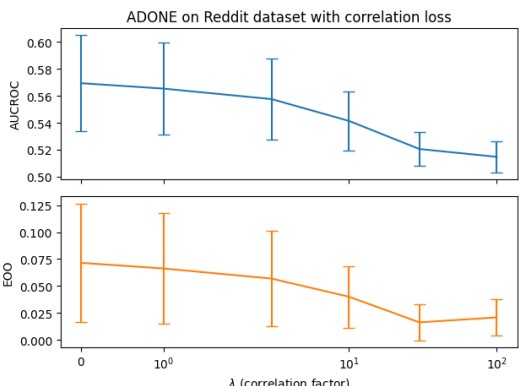

Figure XXII: Accuracy (AUCROC, left, higher is better) and Fairness (Equality of Odds, right, lower is better) metric for AdONE with the correlation regularizer on the Reddit dataset.

The findings indicate that for GAD methods like CONAD, DOMINANT, and VGOD, which achieve high accuracy, there were minimal changes in AUCROC and EOO. However, we observed that AdONE and DONE are highly sensitive to attribute biases in their encoding, resulting in a decrease in EOO. Furthermore, for COLA, we suspect that the sensitive attribute is a prominent indicator of contrast because it employs the contrastive learning technique between neighbors. Nevertheless, it is important to note that these three methods do not perform well in terms of accuracy, and therefore do not justify the significance of eliminating sensitive attributes from the dataset.

# K    IMPLEMENTATION DETAILS

All the experiments were conducted with the NVIDIA DGX-1 system with 8 NVIDIA TESLA V100 GPUs.

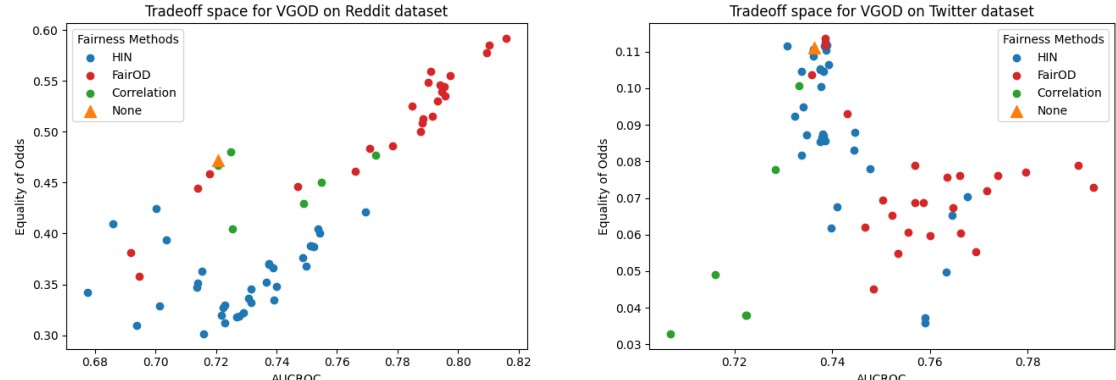

Figure XXIII: Trade-off space for VGOD on the Reddit (left) and Twitter (right) datasets.

Table V: The changes in accuracy and unfairness metrics according to the elimination of sensitive attributes.

| Metric | w/ Sensitive Attributes | | w/o Sensitive Attributes | |
|---|---|---|---|---|
| | AUCROC | EOO | AUCROC | EOO |
| CoLA | 0.453±0.014 | 0.177±0.014 | 0.450±0.014 | 0.035±0.024 |
| CONAD | 0.608±0.001 | 0.055±0.002 | 0.609±0.001 | 0.056±0.003 |
| DOMINANT | 0.608±0.001 | 0.057±0.003 | 0.608±0.001 | 0.058±0.003 |
| VGOD | 0.721±0.009 | 0.472±0.063 | 0.721±0.010 | 0.471±0.064 |
| DONE | 0.578±0.033 | 0.068±0.043 | 0.553±0.008 | 0.015±0.007 |
| AdONE | 0.575±0.027 | 0.077±0.048 | 0.554±0.011 | 0.020±0.009 |

For the GAD methods (*i.e.*, CoLA, CONAD, and DOMINANT), we used the default hyperparameters provided by PyGOD. Batch sampling was used for larger datasets, such as our Twitter dataset and its debiased versions after running the graph debiasers (*i.e.*, FAIRWALK and EDITS), with a batch size of 16,384.

For FAIRWALK, the GitHub implementation[18] was used with hyperparameters of hidden dimensions=64, walk length=30, number of walks=200, window size=10, and node batch=4. For EDITS, the GitHub implementation[19] was used with hyperparameters of epoch=500, and learning rate=0.001. Note that even with epoch=1, the results of EDITS on the Reddit dataset still yielded a nearly complete graph with 97M edges.

For the fairness regularizers, the hyperparameters $\lambda$ and $\gamma$, which we used for each regularizer, can be found in Appendix E.

---

[18]https://github.com/urielsinger/fairwalk
[19]https://github.com/yushundong/EDITS

