# OpenReview forum: "Towards Fair Graph Anomaly Detection: Problem, New Datasets, and Evaluation"
_ICLR.cc/2024/Conference — Submitted to ICLR 2024_

### Official Review · Reviewer_rPbE · 2023-10-27

**Soundness:** 3 good
**Presentation:** 4 excellent
**Contribution:** 4 excellent
**Rating:** 8
**Confidence:** 4

**Summary:**

This work tackles the novel fair graph anomaly detection problem. The authors present two graph datasets constructed from Reddit and Twitter. They investigate the performance-fairness trade-off in nine existing GAD and non-graph anomaly detection methods on these datasets with extensive experiments. The results are impressive and demonstrate the effectiveness of the proposed approach.

**Strengths:**

1. The paper is well-written, structured and easy to follow. The authors provide sufficient implementation and experimental details in the Appendix. The authors provide a clear motivation for the problem.
2. The graph datasets constructed from Reddit and Twitter are contribution to the field.
3. The experiments are extensive and the results are presented in a clear and concise manner. The discussion of future directions is a plus.

**Weaknesses:**

1. The paper could benefit from a more detailed and faithful discussion of the limitations of the proposed approach. The authors only mention that the approaches they examined leverage unsupervised learning but not semi-supervised methods, which is not deemed as a “limitation.”
2. Additionally, the paper could benefit from a more detailed discussion of the implications of the results for real-world applications.
3. It would be great if you provide more details in the captions of Figure 1-3. For example, in Figure 1 and 2, “increasing $\lambda$ leads to a decrease in EOO. ” Then what is its implication? You should give more intuitive information about what these plots suggest.
4. What are the implications of the results for real-world applications of anomaly detection in social networks?
5. The datasets proposed are indeed a plus. What is the plan for releasing the dataset? How do you ensure the long-term accessibility of the dataset to a wide range of users?

Minor point: It is a nice practice to also use **bold** subtitles in the first section for better clarity. You did this for the other sections, but can also do it for the introduction.

**Questions:**

See weaknesses.

---

> ### Author Response · Authors · 2023-11-20
> **Author Response to Reviewer rPbE**
>
> We thank the reviewer for her/his careful reading and helpful comments. The following are our responses.
>
> **R1) The paper could benefit from a more detailed and faithful discussion of the limitations of the proposed approach. The authors only mention that the approaches they examined leverage unsupervised learning but not semi-supervised methods, which is not deemed as a “limitation.”**
>
> We apologize for the confusion. However, we first would like to clarify that in Section 5, we discussed TWO promising directions for semi-supervised learning and a possible combination of graph debiasers and fairness regularizers.
>
> Nevertheless, to thoroughly address the reviewer’s concern, we have included more discussion in the revised paper: *Our datasets can be easily extended in semi-supervised learning scenarios once the literature increasingly covers the supervised or semi-supervised GAD methods. In such cases, the only extra tasks are to analyze the parameters for the amount of labels given to the model as well as the ratio of labels pertaining to sensitive attributes.*
>
> The revised paper also discusses the significance of timestamp information in the dynamic GAD process: *Lastly, by infusing this temporal aspect, our datasets will be better able to accommodate the evolution of our problem beyond the confines of a static framework. For instance, we can shed light on critical junctures where fairness considerations may become more pronounced in the context of evolving graph structures, e.g., instances where fairness in GAD methods degrades.*
>
> We have clarified the above points in the revised version of our paper (see page 9).
>
> **R2) Additionally, the paper could benefit from a more detailed discussion of the implications of the results for real-world applications. What are the implications of the results for real-world applications of anomaly detection in social networks?**
>
> We appreciate this suggestion. These findings suggest that identifying anomalies stemming from minority groups poses a challenge when it comes to unsupervised learning. In particular, detecting misinformation from users who lean towards right-leaning political affiliation is more difficult in the current scenario. Consequently, political bias in the FairGAD can result in problems such as the reinforcement of confirmation biases, unequal treatment of news sources, and hindrances in achieving just and precise categorization. We hope that our datasets can encourage further research in this area to enhance detection performance for minority groups.
>
> We have clarified the above points in the revised version of our paper (see page 9).
>
> **R3) It would be great if you provide more details in the captions of Figure 1-3. For example, in Figure 1 and 2, “increasing \lambda leads to a decrease in EOO. ” Then what is its implication? You should give more intuitive information about what these plots suggest.**
>
> We would like to clarify that the sentence mentioned by the reviewer (e.g., increasing \lambda leads to a decrease in EOO) is a part of the main text and not a caption. It should be noted that due to space limitations, we have included the intuitive information and implications in the main text instead of the captions.
>
> **R4) The datasets proposed are indeed a plus. What is the plan for releasing the dataset? How do you ensure the long-term accessibility of the dataset to a wide range of users?**
>
> We thank the reviewer for this careful comment. Upon acceptance of the paper, we will upload our datasets to both Github and Zenodo (https://zenodo.org/) for long-term access and preservation. Furthermore, we will update our current code at the Anonymous Github repository before releasing it on Github. We anticipate users to provide feedback via email or Github issues, and we will version all changes on Zenodo as necessary.
>
> **R5) Minor point: It is a nice practice to also use bold subtitles in the first section for better clarity. You did this for the other sections, but can also do it for the introduction.**
>
> Thank you for your suggestions regarding the clarity and style of our paper. We have incorporated bold subtitles in the first section to improve coherence and readability.

---

### Official Review · Reviewer_jyni · 2023-10-30

**Soundness:** 3 good
**Presentation:** 3 good
**Contribution:** 3 good
**Rating:** 6
**Confidence:** 3

**Summary:**

This paper focuses on the fairness of unsupervised graph anomaly detection (GAD). The background is that fairness in GAD is vital yet under-explored in research, but there is a lack of real-world datasets containing graph structures, anomaly labels, and sensitive attributes for the research. To handle the issue, this paper builds two graph datasets for fair GAD. Besides, it also conducts empirical evaluation: (1) investigating the effectiveness of the nine GAD methods w.r.t accuracy and fairness, and (2) exploring the performance of some fairness methods on the GAD methods. The codes and datasets are publicly available.

**Strengths:**

The contribution of this paper is significant. It customized two real-world GAD datasets for fairness concerns. I believe the datasets would inspire future work.

This work conducted extensive experiments to analyze the datasets and GAD methods. Specifically, it presents detection effectiveness and comprehensive analysis of fairness, e.g., the analysis of accuracy-fairness trade-off.

The whole paper is generally of great readability and well-organized.

**Weaknesses:**

1. I suggest the authors use case studies to introduce more about the datasets, illustrate the practical meanings of graph anomalies on the two datasets, and show why it is important to care about fairness.

2. This paper is about GAD on attributed graphs, but it only considers the fairness of unsupervised GAD methods, ignoring the semi-supervised ones. Hence, it shows limited impacts.

3. The adopted GAD methods are unsuitable. First, it does not involve the recently proposed GAD methods that were proposed in 2023 [1, 2]. Hence, it is unsuitable to claim that SOTA GAD methods are exploited. Second, the adopted three GAD methods are not diverse, i.e., both DOMINAT and CONAD are reconstruction-based methods and quite similar. Please add more GAD methods with diverse working mechanisms, e.g., community-analysis methods [3-5]. It is also suggested to add non-deep learning GAD methods, e.g., Radar [6].

4. The results in Table 2 seem weird. Why does the EDITS significantly boost the detection performance of CONAD and DOMINAT while showing trivial impacts on CoLA? Specifically, the results on CONAD and DOMINAT may indicate that the sensitive attributes (political leaning, gender, and age) are closely related to the “anomalies”. Intuitively, the performance of CoLA (Table 2) and other GAD methods, e.g., VAE and ECOD (Table III), should also be boosted since EDITS changes the characteristics of the graphs.

5. Although Table 1 partly summarizes the related works, I suggest the authors add a related work section and briefly summarize the mainstream methods about fairness in GNNs.

6. Please check the correctness of the reference information. For example, the paper of the GAD method CoLA was published in 2021 rather than 2022. The paper "Contrastive Attributed Network Anomaly Detection with Data Augmentation" has been cited twice.

References:
[1] Huang Y, Wang L, Zhang F, et al. Unsupervised graph outlier detection: Problem revisit, new insight, and superior method, 2023 IEEE 39th International Conference on Data Engineering (ICDE). IEEE, 2023: 2565-2578.
[2] Duan, Jingcan, et al. "Graph anomaly detection via multi-scale contrastive learning networks with augmented view." Proceedings of the AAAI Conference on Artificial Intelligence. Vol. 37. No. 6. 2023.
[3] Duan, Jingcan, et al. "ARISE: Graph Anomaly Detection on Attributed Networks via Substructure Awareness." IEEE Transactions on Neural Networks and Learning Systems (2023).
[4] Zhou S, Tan Q, Xu Z, et al. Subtractive aggregation for attributed network anomaly detection, Proceedings of the 30th ACM International Conference on Information & Knowledge Management. 2021: 3672-3676.
[5] Gutiérrez-Gómez L, Bovet A, Delvenne J C. Multi-scale anomaly detection on attributed networks, Proceedings of the AAAI conference on artificial intelligence. 2020, 34(01): 678-685.
[6] Li J, Dani H, Hu X, et al. Radar: Residual analysis for anomaly detection in attributed networks, IJCAI. 2017, 17: 2152-2158.

**Questions:**

I wonder whether the fairness concern can be eliminated by directly removing the sensitive attributes (political leaning, gender, and age) while using the remaining attributes for GAD model learning. Because the “anomaly” on the two datasets represents whether a user is a real-news or misinformation spreader, which is purely determined by the “correctness of the news’ content”. Removing the sensitive attributes may not affect detecting “anomaly”, e.g., fake-news spreader.

---

> ### Author Response · Authors · 2023-11-20
> **Author Response (1) to Reviewer jyni**
>
> We thank the reviewer for her/his careful reading and helpful comments. The following are our responses.
>
> **R1) I suggest the authors use case studies to introduce more about the datasets, illustrate the practical meanings of graph anomalies on the two datasets, and show why it is important to care about fairness.**
>
> We appreciate this helpful suggestion. Following the reviewer’s suggestion, we have added the practical meanings and importance of our datasets.
>
> Addressing concerns about fairness in politically biased misinformation detection has significant practical implications. Specifically, political bias can result in problems such as the reinforcement of confirmation biases, unequal treatment of news sources, and hindrances in achieving just and precise categorization. In this case, misclassifying minority groups as misinformation spreaders (i.e., anomalies) can amplify biases and stereotypes and potentially reinforce existing divisions through algorithmic misuse. Therefore, prioritizing fairness builds trust in the detection process and fosters a more equitable information environment. This strategy not only alleviates unintentional consequences, but also facilitates principled and ethical application of misinformation detection systems. This promotes a stronger information network for various user populations. We hope that our datasets can encourage further research in this area to enhance detection performance for minority groups.
>
> We have clarified the above points in the revised version of our paper (see page 4).
>
> **R2) This paper is about GAD on attributed graphs, but it only considers the fairness of unsupervised GAD methods, ignoring the semi-supervised ones. Hence, it shows limited impacts.**
>
> We appreciate the reviewer’s feedback. In response to the reviewer’s suggestion of exploring a semi-supervised setting, we would like to clarify that the majority of existing GAD research originated as an unsupervised task [1, 2, 3]. The primary goal of our study is to investigate fairness within the established paradigms of these existing studies. Consequently, the extension of our approach to encompass a semi-supervised setting is not a trivial endeavor, and we believe that it deviates from our scope.
>
> We sincerely hope that the reviewer understands our work in this context.
>
> [1] Noble CC, Cook DJ. Graph-based anomaly detection. ACM SIGKDD 2003.
>
> [2] Akoglu, L., Tong, H. & Koutra, D. Graph based anomaly detection and description: a survey. Data Mining and Knowledge Discovery (2015). https://doi.org/10.1007/s10618-014-0365-y
>
> [3] H. Kim, B. S. Lee, W. -Y. Shin and S. Lim, "Graph Anomaly Detection With Graph Neural Networks: Current Status and Challenges," in IEEE Access, vol. 10, pp. 111820-111829, 2022, doi: 10.1109/ACCESS.2022.3211306.
>
> **R3) The results in Table 2 seem weird. Why does the EDITS significantly boost the detection performance of CONAD and DOMINANT while showing trivial impacts on CoLA? Specifically, the results on CONAD and DOMINANT may indicate that the sensitive attributes (political leaning, gender, and age) are closely related to the “anomalies”. Intuitively, the performance of CoLA (Table 2) and other GAD methods, e.g., VAE and ECOD (Table III), should also be boosted since EDITS changes the characteristics of the graphs.**
>
> We thank the reviewer for this careful review.
>
> It should be noted that the key differences between CONAD/DOMINAT and CoLA lie in their utilization of loss functions. First, CONAD and DOMINANT heavily rely on reconstruction error, fully exploiting the augmented graph structure obtained from EDITS. Conversely, CoLA only partially exploits the augmented graph structure by sampling node pairs through random walks. This explains why the EDITs significantly enhance the accuracy of CONAD and DOMINAT, but only slightly improve that of CoLA. We have clarified the above points in the revised version of our paper (see page 7).
>
> Furthermore, we would like to clarify that both VAE and ECOD are non-graph anomaly detection methods. Therefore, any modifications made to the graph structure by EDITS will not be impacted by these methods. While there may still be some changes to node attributes through the use of EDITS, its main goal is to minimize differences in overall attribute distributions. Consequently, the impact of using VAE and ECOD that rely on attribute distributions should be minimal.
>
> **R4) Please check the correctness of the reference information. For example, the paper of the GAD method CoLA was published in 2021 rather than 2022. The paper "Contrastive Attributed Network Anomaly Detection with Data Augmentation" has been cited twice.**
>
> Thank you for identifying these issues; we have fixed them in the revised paper.

---

> ### Author Response · Authors · 2023-11-20
> **Author Response (2) to Reviewer jyni**
>
> **R5) The adopted GAD methods are unsuitable. First, it does not involve the recently proposed GAD methods that were proposed in 2023 [1, 2]. Hence, it is unsuitable to claim that SOTA GAD methods are exploited. Second, the adopted three GAD methods are not diverse, i.e., both DOMINANT and CONAD are reconstruction-based methods and quite similar. Please add more GAD methods with diverse working mechanisms, e.g., community-analysis methods [3-5]. It is also suggested to add non-deep learning GAD methods, e.g., Radar [6].**
>
> We appreciate the reviewer’s detailed suggestions and the helpful paper references. To address the reviewer’s suggestion, we performed further experiments with two GAD methods: (1) VGOD [1], which is a SOTA GAD method proposed in 2023; and (2) ONE [2], which is a non-deep learning GAD method based on matrix factorization.
>
> [1] Huang Y, Wang L, Zhang F, et al. Unsupervised graph outlier detection: Problem revisit, new insight, and superior method, 2023 IEEE 39th International Conference on Data Engineering (ICDE). IEEE, 2023: 2565-2578.
>
> [2] Bandyopadhyay, S., Lokesh, N. and Murty, M.N., 2019, July. Outlier aware network embedding for attributed networks. In Proceedings of the AAAI conference on artificial intelligence (AAAI).
>
>
> The results are shown below (see Appendix I for more details):
>
> - Twitter
> | Debiaser	| None	|None|	EDITS|	EDITS|	FairWalk|	FairWalk|
> |-----------|-----------|-----------|-----------|-----------|-----------|-----------|
> |**Metric**|	**AUCROC**|	**EOO**|	**AUCROC**|	**EOO**|	**AUCROC**|	**EOO**|
> |**VGOD**|0.736±0.006|0.111±0.021|0.823±0.032|0.144±0.085|0.602±0.003|0.052±0.004|
> |**ONE**|0.501±0.005|0.010±0.008|0.501±0.005|0.010±0.008|0.544±0.005|0.025±0.011|
>
> - Reddit
> | Debiaser	| None	|None|	EDITS|	EDITS|	FairWalk|	FairWalk|
> |-----------|-----------|-----------|-----------|-----------|-----------|-----------|
> |**Metric**|	**AUCROC**|	**EOO**|	**AUCROC**|	**EOO**|	**AUCROC**|	**EOO**|
> |**VGOD**|0.721±0.009|0.472±0.063|o.o.m|o.o.m|0.673±0.002|0.295±0.006|
> |**ONE**|0.496±0.007|0.014±0.009|o.o.m|o.o.m|0.524±0.008|0.035±0.021|
>
> As shown in the table above, ONE is not effective in this problem due to the challenge of capturing nonlinear relationships using the matrix factorization techniques. Additionally, ONE assigns equal weight to all node attributes, which may not be necessary given the large number of attributes for each node. On the other hand, VGOD performs very well in both datasets. The main contribution of VGOD stems from its capacity to capture neighborhood variance as a determining factor in identifying structural outliers. This is achieved by sampling both positive and negative edge sets, while attribute outliers are handled in a standard manner comparable to other GAD methods by reconstructing the node attributes. As such, the improved accuracy (AUCROC) can be attributed to the detection of more structural outliers. However, this also leads to an increase in unfairness (EOO), which is not desirable in FairGAD. We posit that this is because of the significant discriminatory effect of VGOD in identifying neighborhood variances. As a result, structural outliers of nodes with the majority sensitive attribute are more likely to be identified since they form a larger proportion of both positive and negative edges.
>
> We have included the above results in Appendix I. We conducted further investigation of the trade-off space for VGOD on our Reddit and Twitter datasets. Please refer to Appendix I for the detailed results and discussions because the figures illustrating the results cannot be uploaded on Openreview.
>
> **R6) Although Table 1 partly summarizes the related works, I suggest the authors add a related work section and briefly summarize the mainstream methods about fairness in GNNs.**
>
> We appreciate the suggestion. The related works for our paper are classified into three categories: (1) graph anomaly detection (GAD), (2) fair graph mining, and (3) relevant datasets w.r.t graph, anomaly detections, and fairness. We agree that it would be better to summarize the related works in a single section. Nevertheless, due to space limitations, we had to separate the summary for relevant papers, instead of including a related work section. In Section 1, we presented the existing relevant datasets and their limitations in the context of the FairGAD problem. In Section 4.1, we provided an overview of the existing GAD methods, including DOMINANT, CONAD, and CoLA. We also discussed the fair graph mining methods, such as FairWalk (for non-GNNs) and EDITS (for GNNs); here, it should be noted that the literature includes both non-GNNs and GNNs-based GAD methods, so we should explore the fairness in both scenarios.
>
> In this context, we believe that this paper offers a concise review of all relevant works.

---

> ### Author Response · Authors · 2023-11-20
> **Author Response (3) to Reviewer jyni**
>
> **R7) I wonder whether the fairness concern can be eliminated by directly removing the sensitive attributes (political leaning, gender, and age) while using the remaining attributes for GAD model learning. Because the “anomaly” on the two datasets represents whether a user is a real-news or misinformation spreader, which is purely determined by the “correctness of the news’ content”. Removing the sensitive attributes may not affect detecting “anomaly”, e.g., fake-news spreader.**
>
> We thank the insightful comment.
>
> First, simply removing sensitive attributes is inadequate since the literature [1, 2] suggests that these attributes are already embedded in the graph structure of the dataset. For example, users with the same sensitive attributes tend to have more connections than those with differing sensitive attributes. Furthermore, it is important to clarify that our analysis relies on unsupervised learning since GAD is mostly considered an unsupervised problem in the literature. Thus, no particular emphasis or weight is given to sensitive attributes when detecting anomalies. Especially considering that the sensitive attribute is only one out of 385 or 780 attributes within the node attributes, we believe that it does not significantly impact anomaly detection.
>
> [1] Yu Wang, Yuying Zhao, Yushun Dong, Huiyuan Chen, Jundong Li, Tyler Derr, Improving Fairness in Graph Neural Networks via Mitigating Sensitive Attribute Leakage. KDD 2022: 1938-1948.
>
> [2] Yushun Dong, Ninghao Liu, Brian Jalaian, Jundong Li, EDITS: Modeling and Mitigating Data Bias for Graph Neural Networks. WWW 2022: 1259-1269.

---

> ### Comment · Reviewer_jyni · 2023-11-22
>
> I have read the responses from the authors. I maintain my rating. The responses addressed some of my concerns.
>
> Some of my questions have been answered in a general way. For example, specific examples cannot be given to illustrate the practical meanings of emphasizing fairness in graph anomaly detection. But this is a common problem in graph anomaly detection, because the industry is still using lots of rules and annotations to perform graph anomaly detection. My last question, i.e., whether the fairness concern can be eliminated by directly removing the sensitive attributes, can actually be answered by conducting the corresponding experiments.
>
> In summary, this paper released two large datasets. I am not quite sure about the standard of ICLR dataset papers. The workload and technical contributions are less than regular papers with novel frameworks with or without new datasets. The impact of this paper, e.g., whether the graph anomaly detection research community can have new discoveries or not by using the two large datasets, may need some time to evaluate. Thus,  I maintain my rating.

---

> > ### Author Response · Authors · 2023-11-23
> > **Thank you for your feedback**
> >
> > We would like to express our sincere gratitude for your deep engagement and for providing us with invaluable feedback. Also, we are glad to hear that some of your concerns have been addressed.
> >
> > **(1) My last question, i.e., whether the fairness concern can be eliminated by directly removing the sensitive attributes, can actually be answered by conducting the corresponding experiments.**
> >
> > Following the reviewer’s feedback, we additionally conducted experiments that removed the sensitive attribute on the Reddit dataset. The results are below:
> >
> > |	|w/ Sensitive Attributes	|w/ Sensitive Attributes|	w/o Sensitive Attributes|	w/o Sensitive Attributes|
> > |-----------|-----------|-----------|-----------|-----------|
> > |**Metric**|	**AUCROC**|	**EOO**|	**AUCROC**|	**EOO**|
> > |CoLA    | 0.453±0.014 | 0.177±0.014 | 0.450±0.014 | 0.035±0.024        |
> > |CONAD    | 0.608±0.001 | 0.055±0.002 | 0.609±0.001 | 0.056±0.003      |
> > |DOMINANT | 0.608±0.001 | 0.057±0.003 | 0.608±0.001 | 0.058±0.003      |
> > |VGOD | 0.721±0.009 | 0.472±0.063 | 0.721±0.010 | 0.471±0.064      |
> > |DONE    | 0.578±0.033 | 0.068±0.043 | 0.553±0.008 | 0.015±0.007     |
> > |AdONE   | 0.575±0.027 | 0.077±0.048 | 0.554±0.011 | 0.020±0.009     |
> >
> > The findings indicate that for GAD methods like CONAD, DOMINANT, and VGOD, which achieve high accuracy, there were minimal changes in AUCROC and EOO. However, we observed that AdONE and DONE are highly sensitive to attribute biases in their encoding, resulting in a decrease in EOO. Furthermore, for COLA, we suspect that the sensitive attribute is a prominent indicator of contrast because it employs the contrastive learning technique between neighbors. Nevertheless, it is important to note that these three methods do not perform well in terms of accuracy, and therefore do not justify the significance of eliminating sensitive attributes from the dataset.
> >
> > We have clarified the above points in Appendix J. Again, we thank the reviewer for this helpful comment.
> >
> > **(2) In summary, this paper released two large datasets. I am not quite sure about the standard of ICLR dataset papers. The workload and technical contributions are less than regular papers with novel frameworks with or without new datasets. The impact of this paper, e.g., whether the graph anomaly detection research community can have new discoveries or not by using the two large datasets, may need some time to evaluate.**
> >
> > We would like to clarify that the contributions of our paper lie not only in providing new datasets, but also in formulating a new FairGAD problem, conducting a comprehensive analysis of the important characteristics of the datasets in terms of the FairGAD problem, and exploring the performance of GAD methods with/without fairness methods. Therefore, we believe that our contributions as the FIRST paper tailored to the new FairGAD problem are not trivial (i.e., as we are not addressing a benchmark paper under the existing well-known problem), and will also provide valuable insights, including additional data collection in the future.
> >
> > In addition, identifying appropriate datasets for the FairGAD problem is quite challenging as the data needs to be made public, contain a ground truth anomaly, and have sensitive attribute labeling. Obtaining data that meets all these requirements is difficult due to: (i) Lack of Literature: No previous work has explored the collection of these datasets nor provided guidelines on meeting the above requirements; (ii) Ethical Considerations: These labels related to anomalies and sensitive features must be defined carefully to avoid the perpetuation of biases. Different domains, including social media, finance, and email, require exploration of different correlations between anomalies and sensitive features. Such ethical considerations are discussed in the paper, see Section 6. In this context, our contribution is a strong foundation and a valuable resource for the community.
> >
> > We sincerely hope that the reviewer understands our work in this context.

---

### Official Review · Reviewer_9VrT · 2023-10-31

**Soundness:** 2 fair
**Presentation:** 3 good
**Contribution:** 2 fair
**Rating:** 5
**Confidence:** 3

**Summary:**

The paper presents two new datasets to foster fairness research in graph-based anomaly/outlier detection tasks. The first consists of Reddit data (approx. 10k nodes, average degree 122.5) created by connecting users who posted to the same subreddit within a 24-hour window, from a set of 110 politics-related subreddits. The second consists of Twitter data (approx. 48k nodes, average degree 9.8) created from "follower" relationships of a list of users who posted COVID-19 related misinformation (Verma et al., 2022). In the dataset curation, several node features were incorporated, including demographic attributes and political leaning. Labels were defined based on whether the user posted misinformation. These and other datasets lacking one of "graph", "anomaly detection" and "fairness" aspects are characterized. The authors formalize the FairGAD problem -- fair graph anomaly detection -- and propose performance (AUC and AUPR) and fairness metrics (SP, EOO) for evaluation. Using the new datasets, they evaluate three GAD methods (CoLA, CONAD, DOMINANT) in combination with a graph debiaser (FairWalk or EDITS) or with a fairness regularizer (FairOD, HIN or Correlation).

**Strengths:**

S1. Work includes a great data collection and dataset curation effort (including) data from two major social networks and may help to address the lack of datasets for studying fairness in graph anomaly detection.

S2. The methodology is relatively thorough w.r.t. the choice of GAD and debiasing techniques.

S3. Proper documentation of the dataset using a Datasheet (Appendix A).

S4. The paper is well-written, the reading flows well and the appendices provide extensive results and details.

**Weaknesses:**

W1. Imbalanced classification with sensitive attributes could also be used for evaluating FairGAD methods by discarding labels. The paper below includes an NBA network of basketball players:
- Enyan Dai and Suhang Wang. Say no to the discrimination: Learning fair graph neural networks with limited sensitive attribute information. In WSDM ’21. URL https://doi.org/10.1145/3437963.3441752.

W2. Defining edges in Reddit based on users who posted to the same subreddit within a time window weakens the argument that this is a real graph. This choice bears some resemblance with "synthetic" graphs connecting all nodes that have some common property. Moreover, it creates extremely dense subgraphs which prevented the authors from running the EDITS fairness regularizer. The relatively low performance (AUCROC ~0.61) suggests that the edges may not be encoding useful information.

W3. Although demographic attributes of the users are inferred through M3, they were not considered in the experiments as alternative choices of sensitive attributes (which would seem very natural).  Note that this is part of the motivation in Section 2.

W4. Some parts of the text require clarification/revision.

W5. As raised by the AC, the new datasets require a suitability check.

**Questions:**

Q1. Is there a fundamental issue preventing the use of graph datasets introduced for classification tasks which have both imbalanced labels and sensitive attributes for FairGAD, assuming the labels are not used, or only partially used?

Q2. Have you considered more organic ways of defining edges in the Reddit network? For instance, users who replied to each other?

Q3. Have you considered setting age, gender or race as sensitive attributes? If not, why? If so, do you have preliminary results to comment on?

Q4.  Clarification questions:
- In the intro, "Jin et al., 2023a" does not seem related to cybersecurity.
- In the intro, which reference supports the use of GAD methods in loan applications?
- By "posted to the same subreddit", does that include comments or only submissions? Are you referring to the "same subreddit thread/submission"?
- Is the Reddit network weighted? If not, wouldn't that be important?
- What does Y=0 indicate? Unlabeled or normal?
- Using "Statistical Parity Difference" and "Equal Opportunity Difference" would be preferable since the current metrics (Eqs. 1-2) are "unfairness" metrics.
- In Section 4.2, the authors conjecture that the limited performance may be due to reliance on graph homophily. Can you provide a metric as evidence to back up this hypothesis? Alternatively, is it possible that the edge definition in the Reddit network is not capturing useful information?
- Review: "However, we believe that the gain *of improvement* is not substantial" -> in performance?
- In "none of the existing GAD methods fail to achieve the desired outcomes", isn't the desired outcome low EOO and high AUC?
- In Appendix C, it is not clear whether $A_{norm}$ refers to the symmetric normalized or the random walk graph Laplacian.

**Details Of Ethics Concerns:**

The authors stated that the collection of publicly available datasets was deemed review exempt by the IRB.

---

> ### Author Response · Authors · 2023-11-20
> **Author Response (1) to Reviewer 9VrT**
>
> We thank the reviewer for her/his careful reading and helpful comments. The following are our responses.
>
> **R1) Although demographic attributes of the users are inferred through M3, they were not considered in the experiments as alternative choices of sensitive attributes (which would seem very natural). Note that this is part of the motivation in Section 2. Have you considered setting age, gender or race as sensitive attributes? If not, why? If so, do you have preliminary results to comment on?**
>
> We appreciate the reviewer’s feedback. However, we would like to clarify that we have already conducted some preliminary experiments that consider the age and gender of users, as inferred by the M3 system, as alternative sensitive attributes. Please see Appendix G for further details.
>
> For your convenience, please find some results below:
> |Sensitive Attribute|Political Leaning|Gender|Age|
> |-----------|-----------|-----------|-----------|
> |CoLA|0.023±0.012|0.007±0.004|0.017±0.006|
> |CONAD|0.044±0.003|0.030±0.004|0.036±0.007|
> |DOMINANT|0.044±0.003|0.031±0.004|0.037±0.007|
>
> This table shows the different values of the fairness metric (i.e., Equality of Odds (EOO), lower is better) as the sensitive attribute changes. Note that the accuracy, as measured by AUCROC, remains relatively constant because the node attributes and the network structure remain unchanged throughout the shifts in the sensitive attribute. On the other hand, fairness, as measured by EOO in this table, shows the tendency for different levels of unfairness to manifest across different sensitive attributes. By choosing the user's political leanings as the sensitive attribute, a Twitter dataset with increased levels of unfairness was created. Thus, we believe it would be easier to analyze the difference in fairness metrics after applying fairness regularizers or graph debiasers.
>
> We conducted further investigation into how modifying the sensitive attribute affects the fairness regularizers. Please refer to Appendix G for the detailed results and discussions because the figures illustrating the results cannot be uploaded on Openreview.
>
> **R2) Imbalanced classification with sensitive attributes could also be used for evaluating FairGAD methods by discarding labels. The paper below includes an NBA network of basketball players: Enyan Dai and Suhang Wang. Say no to the discrimination: Learning fair graph neural networks with limited sensitive attribute information. In WSDM ’21. URL https://doi.org/10.1145/3437963.3441752. Is there a fundamental issue preventing the use of graph datasets introduced for classification tasks which have both imbalanced labels and sensitive attributes for FairGAD, assuming the labels are not used, or only partially used?**
>
> Thank you for the reviewer’s advice.
>
> However, it should be noted that the datasets for the FairGAD problem possess real graph structure, real anomaly labels, and real sensitive attributes. While we recognize the paper referenced by the reviewer, which includes the NBA network, the dataset cannot be utilized in addressing the FairGAD problem. This is because the labels present in the dataset indicate the nationality of basketball players and do not hold any relevance to anomalies (i.e., missing ground truth).
>
> In this context, we have created two datasets for the first time with real graph structure, real anomaly labels, and real sensitive attributes.

---

> ### Author Response · Authors · 2023-11-20
> **Author Response (2) to Reviewer 9VrT**
>
> **R3) Defining edges in Reddit based on users who posted to the same subreddit within a time window weakens the argument that this is a real graph. This choice bears some resemblance with "synthetic" graphs connecting all nodes that have some common property. Moreover, it creates extremely dense subgraphs which prevented the authors from running the EDITS fairness regularizer. The relatively low performance (AUCROC ~0.61) suggests that the edges may not be encoding useful information. Have you considered more organic ways of defining edges in the Reddit network? For instance, users who replied to each other?**
>
> We thank the reviewer for the insightful comments.
>
> However, we believe that our dataset possesses an “organic” graph structure due to its origins in actual user behavior. The graph structure in the Reddit dataset stems from connecting two users who have interacted in the same subreddit in the past. This design choice was influenced by previous research findings, which suggest that users who interact within the same online community with close temporal proximity are likely to be aware of each other's posts or share similar topical interests [1, 2, 3]. In contrast, the graph structures of existing datasets are created based solely on the Minkowski distance between node attributes, without any consideration of actual user behavior. In this context, we believe that the graph structure in the Reddit dataset is constructed in a more organic manner compared to existing datasets.
>
> Furthermore, we would like to clarify that the density of the graph in the Reddit dataset is approximately 1.24% (=(1,211,748/(9,892*9,892)*100)), indicating that it is not highly dense. In comparison, the density of the existing German dataset is approximately 2.45%(=(24,970/(1,000*1,000)*100)).
>
> We have clarified the above points in the revised paper (see page 5).
>
> [1] Krohn, Rachel, and Tim Weninger. "Subreddit Links Drive Community Creation and User Engagement on Reddit." Proceedings of the International AAAI Conference on Web and Social Media. Vol. 16. 2022.
>
> [2] Waller, Isaac, and Ashton Anderson. "Generalists and specialists: Using community embeddings to quantify activity diversity in online platforms." The World Wide Web Conference. 2019.
>
> [3] Weerasinghe, Janith, Rhia Singh, and Rachel Greenstadt. "Using Authorship Verification to Mitigate Abuse in Online Communities." Proceedings of the International AAAI Conference on Web and Social Media. Vol. 16. 2022.
>
> **R4) Some parts of the text require clarification/revision.**
>
> We appreciate this suggestion. We have carefully proofread and revised our manuscript to enhance its linguistic quality and clarity.
>
> **R5) Clarification questions.**
>
> We appreciate the reviewer’s helpful feedback. We have addressed these questions in the revised paper as follows:
>
> >*In the intro, "Jin et al., 2023a" does not seem related to cybersecurity. In the intro, which reference supports the use of GAD methods in loan applications?*
>
> Thanks for the kind correction. We have updated the references in the introduction section (see page 1).
>
> >*By "posted to the same subreddit", does that include comments or only submissions? Are you referring to the "same subreddit thread/submission"?*
>
> The Reddit dataset includes only submissions without any comments, and we link users by an edge if they posted to the same subreddit within 24h.
>
> >*Is the Reddit network weighted? If not, wouldn't that be important?*
>
> The Reddit network is not weighted. This is because we are linking users if they have posted to the same subreddit within 24 hours, thus this will not take into account the reddit upvotes/downvotes of comments/posts.
>
> >*What does Y=0 indicate? Unlabeled or normal?*
>
> A value of 0 indicates that the node is normal.
>
> >*Using "Statistical Parity Difference" and "Equal Opportunity Difference" would be preferable since the current metrics (Eqs. 1-2) are "unfairness" metrics.*
>
> Thanks for the feedback. We have adjusted the ‘fairness metrics’ to ‘unfairness metrics’ (see page 3).

---

> ### Author Response · Authors · 2023-11-20
> **Author Response (3) to Reviewer 9VrT**
>
> >*In Section 4.2, the authors conjecture that the limited performance may be due to reliance on graph homophily. Can you provide a metric as evidence to back up this hypothesis? Alternatively, is it possible that the edge definition in the Reddit network is not capturing useful information?*
>
> Thank you for the reviewer’s feedback. We acknowledge that GAD methods may have limited performance due to various factors. Nevertheless, we deduced that their prevalent reliance on graph homophily could be a contributing factor based on the following observations. Firstly, GAD methods on attribute graphs heavily depend on both graph homophily and node attributes. In this case, our observations indicate that our datasets generally exhibit a lower degree of structural bias than existing datasets. Conversely, the attribute bias in the Reddit dataset is greater than in Twitter and existing datasets. Furthermore, we examined various graph properties, such as the Gini coefficient and Entropy, in both our datasets and existing datasets. Based on our findings, we have confirmed that our datasets possess properties similar to the existing datasets (refer to Table I in the Appendix). Considering such observations, we attribute the results of low accuracy to GAD methods to low structural bias, which indicates that the graph homophily does not hold significantly. While we acknowledge that unknown factors may have an impact on this phenomenon, it remains outside the scope of our paper; we leave this as a future study.
>
> >*Review: "However, we believe that the gain of improvement is not substantial" -> in performance?*
>
> Thanks for the careful review. We have clarified the relevant sentences as follows (see page ): “Therefore, the results indicate that we can achieve improvements in both performance and fairness by appropriately setting the values of $\gamma$. However, we believe that the gain of improvement is not substantial in either metric.”
>
> >*In "none of the existing GAD methods fail to achieve the desired outcomes", isn't the desired outcome low EOO and high AUC?*
>
> An ideal FairGAD method should attain high AUCROC and low EOO, which would position it in the bottom right corner of Figure 3. However, GAD methods align with a straight line, demonstrating a linear trade-off between performance and fairness even after applying the fairness methods.
>
> >*In Appendix C, it is not clear whether A_norm refers to the symmetric normalized or the random walk graph Laplacian.*
>
> The symmetric normalized adjacency matrix is represented by A_norm (see page 22).

---

> > ### Comment · Reviewer_9VrT · 2023-11-21
> >
> > I am very satisfied with the responses provided by the authors to me and to the other reviewers. In particular, I can single out the additional experiments involving state-of-the-art GAD methods and the alternative sensitive attributes. Therefore, I have updated my rating accordingly.
> >
> > As a minor comment, in their answer to my clarification request
> > > In "none of the existing GAD methods fail to achieve the desired outcomes", isn't the desired outcome low EOO and high AUC?
> >
> > they seem to agree that "none of the existing GAD **achieve** the desired outcome (i.e., bottom right corner)" or, equivalently, that "the evaluated GAD methods **fail** to achieve the desired outcome", which is in contradiction with the current statement.

---

> > > ### Author Response · Authors · 2023-11-21
> > > **Thank you for your feedback**
> > >
> > > We are pleased to know that our responses addressed your concerns. We sincerely thank you for raising the rating of our paper!
> > >
> > > We also appreciate the correction of the typo. Thanks to your careful review, we have updated the sentence correctly, i.e., *Nevertheless, none of the existing GAD methods achieve the desired outcomes (i.e., bottom right corner)* (see page 9).

---

### Official Review · Reviewer_QKDb · 2023-10-31

**Soundness:** 2 fair
**Presentation:** 3 good
**Contribution:** 2 fair
**Rating:** 3
**Confidence:** 1

**Summary:**

This paper presents FairGrad, two datasets with political leanings as sensitive attributes and misinformation spreaders as anomaly labels investigate the fair graph anomaly detection problem. Their performance analysis suggests a performance-fairness trade-off in nine existing anomaly detection methods on five fairness methods and identify limitations in addressing the fair graph anomaly detection problem.

Overall, this is a nice paper that investigates the performance-fairness tradeoff with new benchmark datasets, existing fairness and GAD methods, and performance and fairness metrics. However, the given two datasets may not be representative enough to study the fair graph anomaly detection problem to make strong conclusions. I propose authors to extend the benchmark data collection to more diverse real world datasets, and introduce different synthetic graph families (e.g., dK random, synthetic attribute graph generation) to study more on the impact of structural and attribute bias in the given problem domain.

**Strengths:**

* Important problem domain to evaluate biased and unfair anomaly detection outcomes
* Experiment with existing fairness and GAD methods.

**Weaknesses:**

* Why not taking the distribution of nodes across sensitive classes in calculating the fairness metrics? For example, we need to be aware that predictions on the minority classes are highly represented and vise-versa.
* Data Coverage: Twitter datasets were collected from a set of authors, who posted COVID-19 related tweets that contain misinformation. Political leaning of users is defined as the sensitive attribute. Two users relate to a directed edge if users follow. Apart from the graph characteristics, how representative the given data sample to study the fair graph anomaly detection problem since the dataset focus on specific event such as COVID-19?
* Not sure whether calculating the structural bias from 2-hop neighborhood information is being the most optimal in this problem scenario. For example, Twitter and Reddit datasets are very different from the network structure but yet the structural bias remains comparable.
> structural bias (Dong et al., 2022) uses the Wasserstein-1 distance (Villani, 2021) while comparing adjacency matrices based on a two-hop neighborhood between them
* Given the definition of structural bias taken into account, it is hard to conclude that the limited performance of GAD methods due to only graph homophily or attribute bias.
> Given that our datasets manifest a lower degree of a structural bias when compared to existing synthetic datasets, the limited performance of GAD methods may be due to their prevalent reliance on graph homophily.
> Considering that the attribute bias of Reddit is significantly larger than that of Twitter while their structural biases are similar (see Table 1), we attribute the results of high SP and EOO on Reddit to its substantial attribute bias.

**Questions:**

* Apart from the common practices, are there any reasons not to consider graph anomaly detection problem as a (semi) supervised task?
> It is worth noting that since GAD is regarded as an unsupervised problem in most literature (Kim et al., 2022; Ma et al., 2021), the labels should only be used in the test step, not in the training step
* Does the sensitive attribute need to be predefined?
> FairGAD methods aim to accurately detect anomalous nodes while avoiding discriminatory predictions against individuals from any specific sensitive group.

**Details Of Ethics Concerns:**

This study presents several social media datasets with sensitive attributes.

---

> ### Author Response · Authors · 2023-11-20
> **Author Response (1) to Reviewer QKDb**
>
> We thank the reviewer for her/his careful reading and helpful comments. The following are our responses.
>
> **R1) Not sure whether calculating the structural bias from 2-hop neighborhood information (from EDITS) is being the most optimal in this problem scenario. For example, Twitter and Reddit datasets are very different from the network structure but yet the structural bias remains comparable.**
>
> We appreciate the reviewer’s thoughtful observation. To address the reviewer’s concern, we analyzed the potential for structural bias based on changes in the k-hop neighborhood information (from EDITS) in our datasets (i.e., Reddit and Twitter), as well as in existing datasets (i.e., German, Credit, and Bail). Please refer to Appendix C for the detailed results because the figures illustrating the results cannot be uploaded on Openreview.
>
> For your convenience, we can briefly summarize the results as follows. In general, bias metrics exhibit a monotonic increase for hops of two or more, potentially due to nodes with similar sensitive attributes being more strongly connected. Consequently, we suggest adhering to the metrics outlined in EDITS for a more accurate characterization of the dataset.
>
> **R2) Why not taking the distribution of nodes across sensitive classes in calculating the fairness metrics? For example, we need to be aware that predictions on the minority classes are highly represented and vise-versa.**
>
> We appreciate the insightful comments. We agree that developing better fairness metrics is a crucial and promising direction for future research. When our datasets become publicly available, they could provide a foundation for advancing in this direction.
>
> However, we would like to clarify that the primary goal of our study is to investigate fairness within the established paradigms of these existing studies. Therefore, we utilized extensively-used metrics in previous fair graph mining research [1, 2, 3, 4, 5], instead of developing novel fairness metrics. We sincerely hope that the reviewer understands our work in this context.
>
> [1] Wang, Yu, et al. "Improving fairness in graph neural networks via mitigating sensitive attribute leakage." Proceedings of the 28th ACM SIGKDD Conference on Knowledge Discovery and Data Mining. 2022.
>
> [2] Agarwal, Chirag, Himabindu Lakkaraju, and Marinka Zitnik. "Towards a unified framework for fair and stable graph representation learning." Uncertainty in Artificial Intelligence. PMLR, 2021.
>
> [3] Dai, Enyan, and Suhang Wang. "Say no to the discrimination: Learning fair graph neural networks with limited sensitive attribute information." Proceedings of the 14th ACM International Conference on Web Search and Data Mining. 2021.
>
> [4] Dong, Yushun, et al. "Edits: Modeling and mitigating data bias for graph neural networks." Proceedings of the ACM Web Conference 2022.
>
> [5] Guo, Zhimeng, et al. "Towards Fair Graph Neural Networks via Graph Counterfactual." Proceedings of the 32nd ACM International Conference on Information and Knowledge Management. 2023.
>
> **R3) Given the definition of structural bias taken into account, it is hard to conclude that the limited performance of GAD methods due to only graph homophily or attribute bias.**
>
> Thank you for the reviewer’s feedback. To address the reviewer’s concern, we toned down our claim in the revised paper because GAD methods may have limited performance due to various factors (see page 6).
>
> Nevertheless, we would like to clarify the rationale for our claim. Specifically, we deduced that their prevalent reliance on graph homophily may play a role in contributing to the issue. This conclusion is drawn from the following observations. Firstly, GAD methods on attribute graphs heavily depend on both graph homophily and node attributes. In this case, our observations indicate that our datasets generally exhibit a lower degree of structural bias than existing datasets. Conversely, the attribute bias in the Reddit dataset is significantly greater than in both Twitter and existing datasets. Furthermore, we examined various graph properties, such as the Gini coefficient and Entropy, in our datasets and existing ones. Based on our findings, we have confirmed that our datasets possess properties similar to the existing datasets (refer to Table I in the Appendix). Considering such observations, we attribute the results of low accuracy to GAD methods to low structural bias, which indicates that the graph homophily does not hold significantly.
>
> While we acknowledge that unknown factors may have an impact on this phenomenon, it remains outside the scope of our paper; we leave this as a future study.

---

> ### Author Response · Authors · 2023-11-20
> **Author Response (2) to Reviewer QKDb**
>
> **R4) The given two datasets may not be representative enough to study the fair graph anomaly detection problem to make strong conclusions. I propose authors to extend the benchmark data collection to more diverse real world datasets, and introduce different synthetic graph families (e.g., dK random, synthetic attribute graph generation) to study more on the impact of structural and attribute bias in the given problem domain.
> Data Coverage: Twitter datasets were collected from a set of authors, who posted COVID-19 related tweets that contain misinformation. Political leaning of users is defined as the sensitive attribute. Two users relate to a directed edge if users follow. Apart from the graph characteristics, how representative the given data sample to study the fair graph anomaly detection problem since the dataset focus on specific event such as COVID-19?**
>
> We appreciate the reviewer’s feedback. We also agree with the reviewer that it is valuable to increase the diversity of datasets as much as possible.
>
> However, we would like to clarify that the contributions of our paper lie not only in providing new datasets, but also in formulating a new FairGAD problem, conducting a comprehensive analysis of the important characteristics of the datasets in terms of the FairGAD problem, and exploring the performance of GAD methods with/without fairness methods. Therefore, we believe that our contributions as the FIRST paper tailored to the new FairGAD problem are not trivial (i.e., as we are not addressing a benchmark paper under the existing well-known problem), and will also provide valuable insights, including additional data collection in the future.
>
> Furthermore, we would like to clearly address the scope of our datasets, specifically with respect to users engaging in discussions about COVID-19 (for the Twitter dataset) and politically-related misinformation (for the Reddit dataset). As such, it is important to recognize that our datasets do not fully represent the broader populations on Twitter and Reddit.
>
> The rationale for selecting these particular topics is based on the extensive research exploring misinformation propagation in the context of COVID-19 and politics [1, 2, 3, 4], e.g.,
>
> - Politics: This is particularly significant given the potential polarization and ideological divisions that may arise from the spread of such misinformation, which can affect public discourse and decision-making.
>
> - COVID-19: Unverified claims or inaccurate information about the virus, prevention methods, and treatments can lead to misguided actions that exacerbate the impact of the pandemic and hinder effective response efforts [1, 3, 4].
>
> Expanding our datasets to include other topics could potentially introduce unexpected or unknown biases. This would make the study complex. Given these considerations, our research has focused on two specific topics with well-established correlations in order to maintain clarity and rigor in our analysis.
>
> We have clarified the above points in Section 6 (see “Representativeness of Our Datasets”).
>
> [1] He et al. Racism is a virus: Anti-Asian hate and counterspeech in social media during the COVID-19 crisis. ASONAM 2021.
>
> [2] Micallef et al. The role of the crowd in countering misinformation: A case study of the COVID-19 infodemic. 2020 IEEE International Conference on Big Data (Big Data).
>
> [3] Caceres et al. The impact of misinformation on the COVID-19 pandemic. AIMS 2022.
>
> [4] Naeem et al. COVID-19 Misinformation Online and Health Literacy: A Brief Overview. International Journal of Environmental Research and Public Health.
>
> **R5) Apart from the common practices, are there any reasons not to consider graph anomaly detection problem as a (semi) supervised task?**
>
> As stated by the reviewer, our study did not incorporate semi-supervised GAD methods as our goal was to investigate fairness within the established paradigms of these existing studies. However, it is important to note that our datasets can be easily extended in semi-supervised learning scenarios and could provide a foundation for advancing in this direction. In such cases, the only extra tasks are to analyze the parameters for the amount of labels given to the model as well as the ratio of labels pertaining to sensitive attributes. We believe that these tasks can only be conducted when the supervised or semi-supervised GAD methods are available, which are beyond the scope of our paper.
>
> We have clarified the above points in the revised paper (see page 9).

---

> ### Author Response · Authors · 2023-11-20
> **Author Response (3) to Reviewer QKDb**
>
> **R6) Does the sensitive attribute need to be predefined?**
>
> We appreciate the reviewer’s thorough review. By predefined, it is understood that the sensitive attribute must be defined at the time of training. According to the literature [1, 2, 3, 4], the original GAD methods do not require pre-definition of the sensitive attribute before training as it does not play a role in their training at all. Instead, if we aim to include fairness regularizers or graph debiasers in order to enhance the fairness of the anomaly detection process, it is crucial to predefine the sensitive attribute before the training process so that the methods can utilize it.
>
> [1] Ding, Kaize, et al. "Deep anomaly detection on attributed networks." Proceedings of the 2019 SIAM International Conference on Data Mining. Society for Industrial and Applied Mathematics, 2019.
>
> [2] Xu, Zhiming, et al. "Contrastive attributed network anomaly detection with data augmentation." Pacific-Asia Conference on Knowledge Discovery and Data Mining. Cham: Springer International Publishing, 2022.
>
> [3] Liu, Yixin, et al. "Anomaly detection on attributed networks via contrastive self-supervised learning." IEEE transactions on neural networks and learning systems 33.6 (2021): 2378-2392.
>
> [4] Huang, Yihong, et al. "Unsupervised graph outlier detection: Problem revisit, new insight, and superior method." 2023 IEEE 39th International Conference on Data Engineering (ICDE). IEEE, 2023.

---

> ### Author Response · Authors · 2023-11-22
> **Kindly requesting feedbacks**
>
> Thank you once again for your insightful comments. We have carefully considered your comments and addressed each of them in our rebuttal. It would be very valuable for us to receive your feedback. If there are still concerns that remain in our work, please let us know and we will be happy to discuss them.

---

### Author Response · Authors · 2023-11-20
**Summary of responses**

We sincerely thank you for taking the time to review our paper and provide valuable feedback. We have carefully considered your suggestions and addressed them in our revised manuscript. Below, we summarize the main strengths and suggestions for improvement raised by the reviewers.

The reviewers noted that the paper (i) is well written (Reviewers 9VrT, jyni, and rPbE), (ii) addresses a novel and important problem (Reviewer QKDb), (iii) creates two unique and meaningful datasets (Reviewers 9VrT, jyni, and rPbE), and (iv) performs comprehensive and in-depth experiments (Reviewers QKDb, 9VrT, jyni, and rPbE).

To improve the paper, reviewers asked for (i) further experiments on alternative sensitive attributes (Reviewer 9VrT) and additional baselines (Reviewer jyni), (ii) further explanation on data coverage (Reviewer QKDb), improved methods of graph construction (Reviewer 9VrT), the practical implications and significance (Reviewer jyni), and related research (Reviewer jyni), and (iii) further discussions of the experimental results (Reviewers QKDb and jyni).

To address the reviewers’ comments, we have taken the following steps:
-  Performed new additional experiments with alternative sensitive attributes such as age and gender, and two new baselines based on a recent GAD method and a non-GNN GAD method.
-  Clarified issues related to the representation of data,
-  Justified our graph construction method,
-  Presented practical implications and significance of our datasets, and
-  Provided a detailed discussion of experimental results for enhanced clarity and rigor.

We have responded to each reviewer's comments individually. In addition, we have substantially revised the paper and appendix in response to the reviewers' comments. The changed or new sentences in the revision are highlighted in red.

We thank you again for your valuable feedback and look forward to discussing it with the reviewers.

Best regards,

Authors of paper 6733

---

### Meta-Review · Area_Chair_xtFg · 2023-12-04

**Metareview:**

I have read all the materials of this paper including the manuscript, appendix, comments, and response. Based on collected information from all reviewers and my personal judgment, I can make the recommendation on this paper, *reject*. No objection from reviewers who participated in the internal discussion was raised against the reject recommendation.

**Key Contribution**

This paper proposed two new datasets for FairGAD. Although the authors mentioned the research question of FairGAD is also one of their contributions, such extension from vanilla version to fair version cannot be regarded as a major contribution, since the authors did not propose any new algorithm to tackle FairGAD. Therefore, I put this paper in the category of dataset and benchmark. Definitely, benchmarks move forward the research of related communities.

**New Datasets**

I would like to call the newly collected datasets as semi-real-world datasets, where the node features and labels are obtained by some mature techniques, rather than directly collection from raw information or human annotation.

One reviewer has a major concern on data coverage, which is not well addressed in the response.

For the new datasets, the authors need to check whether the datasets are suitable for the task of FairGAD. This is the key to make this paper self-standing. Unfortunately, it is missing. Here I provide a reference [1], which talks why a widely used dataset in fairness area, *Compas*, is not suitable for fairness learning.

[1] It's compaslicated: The messy relationship between rai datasets and algorithmic fairness benchmarks.

**Challenges**

The authors fail to illustrate the challenges of collecting the new datasets. In another work, I did not see any novel strategy for data collection or annotation, i.e., this paper does not contribute to the data collection community.

**Impact**

The impact of two datasets is limited based on the current version of this paper. I suggest the authors thinking about bigger impacts for diverse research questions and tasks.

**Justification For Why Not Higher Score:**

This paper proposed two new datasets for FairGAD, but it lacks a suitability check.

**Justification For Why Not Lower Score:**

N/A

---

### Decision · Program_Chairs · 2024-01-16

Reject